# New Applications of Photodynamic Therapy in the Management of Candidiasis

**DOI:** 10.3390/jof7121025

**Published:** 2021-11-29

**Authors:** Carmen Rodríguez-Cerdeira, Erick Martínez-Herrera, Gabriella Fabbrocini, Beatriz Sanchez-Blanco, Adriana López-Barcenas, May EL-Samahy, Eder R. Juárez-Durán, José Luís González-Cespón

**Affiliations:** 1Efficiency, Quality and Costs in Health Services Research Group (EFISALUD), Health Research Institute, SERGAS-UVIGO, 36213 Vigo, Spain; erickmartinez_69@hotmail.com (E.M.-H.); beatriz.sanchez.blanco@sergas.es (B.S.-B.); rarenas97@hotmail.com (E.R.J.-D.); epi@uvigo.es (J.L.G.-C.); 2Dermatology Department, Hospital do Vithas Ntra. Sra. de Fatima, 36206 Vigo, Spain; 3Campus of Vigo, University of Vigo, As Lagoas, 36310 Vigo, Spain; 4European Women’s Dermatologic and Venereologic Society (EWDVS), 36700 Tui, Spain; gafabbro@unina.it (G.F.); dra_lopezbarcenas@yahoo.com.mx (A.L.-B.); may.elsamahy@gmail.com (M.E.-S.); 5Psychodermatology Task Force of the Ibero-Latin American College of Dermatology (CILAD), Buenos Aires 2019, Argentina; 6Research Unit, Regional Hospital of High Specialty of Ixtapaluca, Ixtapaluca 56530, Mexico; 7Postgraduate Studies and Research Section, Higher School of Medicine, National Polytechnic Institute, Ciudad de México 07340, Mexico; 8Department of Dermatology, University of Naples Federico II, 80138 Naples, Italy; 9Dermatology Department (Section Mycology), Manuel Gea González Hospital, Tlalpan, Ciudad de México 14080, Mexico; 10Department of Dermatology, University of Medicine, Ain Schams, Cairo 11591, Egypt

**Keywords:** photosensitizer, nanocarriers, *Candida* spp., genes, antimicrobial photodynamic therapy

## Abstract

The most important aetiological agent of opportunistic mycoses worldwide is *Candida* spp. These yeasts can cause severe infections in the host, which may be fatal. Isolates of *Candida albicans* occur with greater frequency and variable resistance patterns. Photodynamic therapy (PDT) has been recognised as an alternative treatment to kill pathogenic microorganisms. PDT utilises a photosensitizer, which is activated at a specific wavelength and oxygen concentration. Their reaction yields reactive oxygen species that kill the infectious microorganism. A systematic review of new applications of PDT in the management of candidiasis was performed. Of the 222 studies selected for in-depth screening, 84 were included in this study. All the studies reported the antifungal effectiveness, toxicity and dosimetry of treatment with antimicrobial PDT (aPDT) with different photosensitizers against *Candida* spp. The manuscripts that are discussed reveal the breadth of the new applications of aPDT against *Candida* spp., which are resistant to common antifungals. aPDT has superior performance compared to conventional antifungal therapies. With further studies, aPDT should prove valuable in daily clinical practice.

## 1. Introduction

The colonisation of different tissues by *Candida* spp. may lead to an infection facilitated by the endogenous proliferation of the microorganism. Consequences of these pathologies can range from benign localised candidiasis to lethal systemic invasions. The high phenotypic biodiversity of *Candida* spp. and alteration of the host’s immune system underlie the variable behaviour of this opportunistic pathogen [1] and can result in novel and amplified invasive properties. In this infectious process, multiple factors interact with the individual to condition the intensity of the virulence of *Candida* spp. [2]. These factors include adherence to epithelial and endothelial cells, production of hydrolytic enzymes (proteases), dimorphic transition by morphogenetic conversion from yeast to a pathogenic mycelial form (hyphae or germ tubes), antigenic variability, the ability to switch between different cell phenotypes, and adhesion to biological and inert substrates.

It should be noted that the virulence factors of *Candida* spp. are necessary for its survival and colonization in both its localized and systemic forms. A number of quorum-sensing (QS) molecules have been described which affect the morphogenesis process in *Candida* spp. Furthermore, the morphological transition of *Candida* spp. in response to changing environmental conditions represents a means by which the strain adapts to different biological niches [1,2]. Furthermore, every morphotype has its own virulence profile, and each pleomorphic form provides critical functions required for pathogenesis. *Candida* spp. produces extracellular hydrolytic enzymes, among which lipases, phospholipases and secreted aspartyl proteinases (Sap) are most significant in virulence. It is known that Sap proteins contribute to pathogenesis by digesting host cell membranes, and also host immune system molecules which prevent antimicrobial attack by the host. One of the most important keys in the development of oral candidiasis is the adhesion of *Candida* to the oral epithelial cells. The first step towards internalization is adhesion to host cells by adhesins. On the other hand, the knowledge of the role of the various virulence factors of *Candida* spp. during in vivo infection has not yet been fully investigated, and more studies are needed that include the quantification of gene expression [1,2].

Clinically paramount *Candida* spp. include *Candida albicans*, a commensal of the human intestinal tract [3], and *C. krusei*, a yeast commensal [4] that is commonly found in mucous membranes. The increase in antifungal resistance has decreased the efficacy of conventional therapies. Treatments are time-consuming and thus demanding on health care budgets. Additionally, current antifungal drugs only have a limited spectrum of action and toxicity. The high level of drug resistance in these microorganisms has spurred the development of antimicrobial photodynamic therapy (aPDT) as a novel and effective means to inactivate these pathogens [5]; aPDT depends on the interaction between light and a photosensitizer (PS) in the presence of oxygen [6]. The mechanism of action of aPDT is based on the use of energy to excite the PS molecule in order to form species such as toluidine ortho blue, with the subsequent production of reactive oxygen species (ROS) or singlet molecule oxygen (^1^O_2_) (Figure 1). The absorption of light induces a change from the S0 singlet state (GS) to the excited singlet state S1 (ES). The excited PS molecule can initiate transformations that lead to the production of radical forms of ROS, which contribute to cell membrane destruction, enzyme inactivation, receptor dysfunction, and DNA damage. Depending on the structure and biochemical composition of the cell, microorganisms have different sensitivities to oxidative processes. Type I reactions occur mainly in Gram-positive bacteria. Type II reactions occur in Gram-negative bacteria and yeasts that possess a more complex cell wall. The thin purine channels in the cell wall prevent the passage of PS. Toluidine ortho blue is especially involved in type II reactions [7]. PDT involves the administration of PS, which selectively accumulates in certain cells or tissues. When the PS is illuminated in the presence of oxygen with a light of adequate wavelength and sufficient dosage, the photooxidation of biological materials occurs, which proves lethal [8] (Figure 1).

Although few singlet oxygen photosensitizers completely satisfy all of the above-mentioned requirements for effective photosensitization, dyes have traditionally been used as organics that absorb visible light. The most commonly used sensitizers for the generation of singlet oxygen are phenalenone (Φ∆ = 1), pink flare (Φ∆ = 0.75), methylene blue (Φ∆ = 0.5), and more recently, coordination compounds based on ruthenium (II) complexes with ligand-chelating polyazaheterocyclics of the 2,2′-bipyridine or 1,10-phenanthroline type and their derivatives (Φ∆ = 0.2–1), as well as derivatives of porphyrins and phthalocyanines and their metal complexes [5,8,9].

As detailed above, they trigger a photochemical reaction, which leads to cell death. PSs must be able to selectively localise to the target pathogen or tissue, and not damage host cells or tissue. PSs must also be capable of homogeneous distribution within the target tissue in order to exert their effect throughout the treated tissue. Finally, PSs must have pronounced photodynamic activity [9].

## 2. Materials and Methods

This systematic review was undertaken to consolidate new applications of PDT in the management of candidiasis, and is reported in accordance with the PRISMA guidelines [10]. The PubMed and Embase databases were searched for articles on aPDT for both human subjects and laboratory animals, plate cultures, and cell cultures. The following search terms were used: (antimicrobial photodynamic therapy) “[All Fields]” OR (*Candida albicans*) OR (*Candida krusei*) OR (*Candida tropicalis*) OR (*Candida glabrata*) OR (non-albicans) OR (*Candida guilliermondii*) OR (*Candida dubliniensis*) OR (*Candida kefyr*) AND (Photosensitizers) AND (hematoporphyrin and porfimer sodium) OR (cyanines) OR (chlorins) OR (porphyrins derivates)/(photosesitizers) AND (nanoparticles) “[All Fields]” OR (liposomes) OR (dendrimers) OR (gold and silver nanoparticles) OR (chitosan) OR (polymeric nano).

Clinical trials, case reports, and other investigations (animal models and in vitro trials) with *Candida* infections treated with aPDT were included. Revisions and guidelines were also included, as well as manuscripts written in languages other than English. The outcomes were reported individually as *Candida* spp. type, PS type, and characteristics. Additional outcomes were side effects and the use of a light source. Whenever data were not available (NA), this was stated. Additional data on the bibliographic searches can be seen in the PRISMA flow diagram (Figure 2). 

## 3. Results

### 3.1. Nanotechnology for Enhanced Efficiency of aPDT

Lipid or organic formulations are being developed in order to improve PS biodistribution and minimise or avoid side effects. The introduction of nanoparticles does not help improve their use. The main characteristics of the nanocarriers are described in Table 1 [11,12,13,14,15,16,17].

In this section, we describe the main applications of aPDT and describe how the encapsulation of PS in liposomes allows high yields of ^1^O_2_ under illumination [18].

Yang et al. [19] used different *Candida* spp., including *C. albicans* strain SC5314 (ATCC MYA-2876D), *C. krusei* (ATCC 6258), and *C. tropicalis* (ATCC 750). For PDT, chlorine e6 (Ce6) was encapsulated in liposomes also containing the cationic surfactant cetyltrimethyl ammonium bromide (CTAB). The liposomes were formulated using various ratios of dimyristoyl-sn-glycero-phosphatidylcholine (DMPC) and CTAB. The strains that responded best were those of *C. albicans*. This is due to the fact that the cell wall of *C. albicans* has a negative charge, and the components of the liposomes facilitate absorption, thereby improving the efficiency of PDT.

Mlynarczyk et al. [20] demonstrated that the performance of dendrimers was similar to that of cationic antimicrobial peptides. This was attributed to their cationic charge and amphipathic conformation, which was induced. The possibility of modifying dendrimer molecules could be valuable for formulating substances that aid in the diagnosis and treatment of candidiasis.

The silicon phthalocyanine Pc 4 is a PS that has been successful in trials due to its hydrophobic nature. However, its formulation and use can be challenging. To permit easy use, poly(amidoamine) dendrimers are used as structures. Baron et al. [21] demonstrated that encapsulated Pc 4 efficiently generated ROS and that the resulting photoactivation was lethal for *Candida*. These results highlight that the use of nanoparticles as a vehicle eliminates toxicity and increases the potency of the drug.

Maliszewska et al. [22] used reference strains of *C. albicans* ATCC 10231 to evaluate the photosensitising activity of rose bengal in combination with biogenic gold nanoparticles at a concentration of 3 ppm, which were synthesised by the cell-free filtrate of *Penicillium funiculosum* BL1 strain a. Suspensions of *C. albicans* colonies were aliquoted to evaluate the antifungal activity. The most effective reduction in the number of planktonic cells was found after 30 min of light irradiation with an Xe lamp (95.4 J/cm^2^). The findings demonstrated that the use of biogenic gold nanoparticles can inactivate viable *C. albicans* cells growing as floating (planktonic) and adherent (biofilm) cultures, with especially pronounced efficacy when used in combination with rose bengal.

Hesieh et al. [23] used a PS comprising a combination of cationic chitosan/tripolyphosphate nanoparticles to encapsulate phthalocyanine. The findings verified that pseudohyphae can be eliminated and the growth of the *C. tropicalis* strains can be reduced. The study opened a new avenue of therapy in patients with resistance to the currently used antifungal therapy.

Hasanin et al. [24] utilised laser light-irradiated 5,10,15,20-tetrakis(m-hydroxyphenyl)porphyrin (mTHPP) loaded onto an ethylcellulose (EC)/chitosan (Chs) composite. This technique improved the physicochemical and photo-destructive properties of mTHPP for *C. albicans* through the use of the thermal and penetration properties of the therapeutic region of the electromagnetic spectrum, and consequently reduced the resistance of the colonies, favouring their elimination. Tang et al. [25] obtained similar excellent results by combining chitosan with 1-[4-(2-aminoethyl) phenoxy] zinc (II) phthalocyanine (ZnPcN) and quaternising the mixture. This mixture was highly effective for the mitochondria of *C. albicans.*

Sakima et al. [26] studied the influence of encapsulating curcumin in polymeric nanoparticles and evaluated the photodynamic effects in a murine model of oral candidiasis. The findings verified that a day of aPDT treatment was more effective than four days of treatment with nystatin. Importantly, the authors also evaluated cytokeratin 13 and cytokeratin 14 by immunohistochemistry.

Evangelista et al. [27] employed nanoparticles containing zinc phenyl-thio-phthalocyanine and amphotericin B (NC/ZnS4Pc + AMB) within poly(lactide-co-glycolide) (PLGA). The authors demonstrated a cytotoxic effect on *C. albicans* ATCC 90028 cells by aPDT. Table 2 summarises the salient aspects of the aforementioned studies. 

### 3.2. Types of PSs Used in aPDT for Candida spp.

The first two compounds used for aPDT were hemato and porfimer sodium. Although both are good PSs, they have the disadvantage of being difficult to eliminate. The resulting long half-life in the skin results in pronounced photosensitivity [28].

In recent years, new compounds have been developed. These second-generation PSs have better properties [29]. These compounds are derived from a tetrapyrrolic ring with absorption maxima between 660 nm and 850 nm. These wavelengths correspond to red and infrared light, which penetrate more deeply into biological tissues (approximately 20 mm).

Within the PS, second-generation compounds were found: cyanines, chlorins, porphyrins, and porphyrin derivatives. Investigations with PS (cyanines, chlorins, porphyrins derivatives and synthetic and natural dyes) in *Candida* spp. have involved liquid suspensions, plates, cell cultures, and animal models. The most relevant results are listed in Table 3.

#### 3.2.1. Cyanines

Maliszewska et al. [30] used reference strains of *C. albicans* ATCC 10231 to evaluate the photosensitising activity of trisulfonated hydroxyaluminium phthalocyanines (AlPcS3) and bimetallic gold and silver nanoparticles synthesised with the cell-free filtrate of *Trichoderma koningii*. Suspensions of *C. albicans* colonies were aliquoted to evaluate the antifungal activity of laser light alone, of the chemical compounds alone in the dark, and of the chemical compounds in the presence of laser light (12.6–94.5 J/cm^2^). Laser light alone at 94.5 J/cm^2^ significantly decreased yeast viability. When laser light was used together with biogenic silver/gold nanoparticles of sulfonated hydroxyaluminium phthalocyanine, viability was reduced by 83.3 ± 1.6% with a 94.5 J/cm^2^ dose of light.

The number of sulfonate groups in the dihydroxyaluminium phthalocyanines was toxic in the absence of light. However, the antifungal activity of these dyes was higher when disulfonated phthalocyanines (AlPcS2) were used, compared to when they were generated by AlPcS3.

Regarding the use of metal oxide nanoparticles and mesoporous silica nanoparticles used in combination with aPDT, the authors did not find it to be effective against *C. albicans*. This reflected the resistance of *C. albicans* to classic antifungals, such as nystatin and amphotericin B or fluconazole. Azizi et al. [31] tried to establish new therapies to more effectively resolve dermatomycosis. The authors prepared and evaluated 130 samples of standard *C. albicans* suspensions (ATCC 10231). The laser used was a diode laser (A.R.C. laser GmbH, Nurnberg, Germany) at wavelengths of 606 nm and 808 nm. In addition, a control group was created in which PS, laser, and classical drugs were not used. Indocyanine green and methylene blue were used as the PSs. The treatment efficacy was greater with the 808 nm laser and 100 Hz pulse repetition rate plus indocyanine green PS, demonstrating a clear superiority over methylene blue.

Hutnick et al. [21] aimed to disperse the particles of the silicon phthalocyanine PS Pc 4 in pure water, with the goal of developing a polymeric nanoparticle as a treatment for the different types of candidiasis, and to address the problems that exist regarding the formulation of the compound and its safety. The applied polymeric nanoparticles were dendrimers. They were manufactured using an interactive synthetic strategy, which improves drug delivery as their uniform size helps to obtain reproducible and reliable biological data. Poly(amidoamine) dendrimers conjugated with polyethylene glycol dendrimers were synthesised. Pc 4 encapsulation was performed, the concentration of the encapsulation was determined, and the conditions of the organisms and cultures were established. *C. albicans* 9652 and SC5314 isolates were used for *C. albicans* uptake studies using a total of 10 μL of phosphate-buffered saline to resuspend the cell pellet and obtain confocal images. The irradiation characteristics were a wavelength of λ = 670–675 and an intensity of 10 J/cm^2^. The doses were necessary to perform the assay to detect cellular ROS, the conversion of non-fluorescent 2′, 7′-dichlorodihydrofluorescein diacetate (H2DCFDA) to fluorescent 2′, and 7′-dichlorofluorescein (DCF). A clonogenic assay was carried out to quantitatively determine cell viability after treatment with the particles and irradiation. For this determination, an XTT assay was used to measure cellular metabolic activity as an indicator of cell viability, proliferation and cytotoxicity, and a colorimetric test (ATCC) was used to measure the metabolic activity of mitochondrial dehydrogenase. In vitro experiments with strains of *C. albicans* demonstrated that the potency of Pc4 was not hampered by the dendrimer used as a vehicle. Thus, after the photoactivation of Pc4, the generated ROS were able to kill *C. albicans* fungal cells.

Acosta et al. [32] described how chlorin e6 (Ce6) takes advantage of both the beneficial photosensitizer properties of Ce6 and the physical and chemical characteristics of the nanoparticle in the preparation of composites using nanoparticles. Graphene nanoparticles, which are composed of graphene oxide (GO) and the reduced form of GO (rGO), were conjugated to several PSs to enhance their performance in PDT. The conjugation of pristine graphene with PS molecules may result in a more efficient and stable material for PDT. Graphene (present as several layers) combined with Ce6 was used as a PS for an antifungal PDT treatment against *C. albicans* (ATCC 90028), in combination with a red light-emitting diode (LED) array as the photoactivation light source. In this design, the conjugated system of graphene π-electrons improves the performance of Ce6 by yielding electrons and consequently delays photobleaching. Greater performance and effectiveness were obtained for *C. albicans*.

New synthetic techniques, including the exfoliation of graphene with the help of amphiphilic molecules, provide stability and solubility for molecules that are otherwise not soluble. In this study, pristine FLG-Ce6 and Ce6 samples were exposed to the singlet oxygen sensor green reagent and photoactivated for 5 min, 10 min, and 15 min of visible light irradiation at 632 nm and at an incident power of 150 mW. Photoactivation of *C. albicans* in the absence of PS did not modify cell viability. However, survival was reduced by up to 10% when *C. albicans* was exposed to Ce6 and then photoactivated for 15 min with LEDs. Moreover, survival was reduced by over 95% by exposure to FLG-Ce6 and 15 min photoactivation. The FLG-Ce6 hybrid nanomaterial has proven to be markedly more effective in PDT against *C. albicans* compared to the effect of Ce6 alone. Another contributor to the increased ROS generation is the ability of FLG to absorb red and near infrared light. The FLG-Ce6 hybrid nanomaterial can enhance the capacity to generate ROS species compared with Ce6 alone, with no fluorescence during the 15 min irradiation time, suggesting that FLG protects Ce6 from photobleaching. The viability of *C. albicans* was drastically reduced when treated with the FLG-Ce6 hybrid nanomaterial and exposed to red light (632 nm) for 15 min. These results are promising for the development of new applications of PDT. More studies are needed to determine the toxicity of this new generation of nanostructured PSs, such as the hybrid nanomaterial described above.

#### 3.2.2. Chlorins

Hidalgo et al. [33] evaluated cultures of *C. albicans* ATCC 96901 subcultured from vial stocks under aerobic conditions and treated by aPDT at a yeast density of 1 × 10^6^ colony forming units (CFU)/mL. The applied PS was a derivative of chlorine e6 (Photodithazine^®^) (PDZ). It was added to each yeast suspension at a final concentration of 100 µM. Other cells, which also received LED (660 nm) with an output power of 690 mW, were used for 2, 4, and 8 min of irradiation. *C. albicans* was eradicated in both aPDT groups. The degree of photoinactivation depended on the irradiation time. In the group where fluconazole was combined with aPDT, the latter displayed an antagonistic effect in the presence of fluconazole and the delayed inactivation of yeast. Furthermore, the absorption of the fluconazole/methylene blue solution decreased upon irradiation with light.

Chlorine derivatives and second-generation PSs can be used in lower concentrations and with a shorter period of illumination. Absorption occurs more strongly in the red spectrum (650–680 nm), which can penetrate deeper into tissues [34]. PDZ is water-soluble and is stable during storage. In vitro studies have demonstrated that PDZ applied alongside visible light is effective in the inactivation of suspensions of *Candida* spp. clinical isolates. However, there are no published data concerning the inactivation of fluconazole-resistant strains. The efficacy of PDZ-mediated aPDT against fluconazole-resistant *C. albicans* (ATCC 10231) was evaluated in a murine model of oral candidiasis by comparing it with the conventional topical antifungal nystatin. PDZ was used as a PS to evaluate whether this new drug formulation would improve the photodynamic effectiveness. For this purpose, a model LXHLPR09 handheld device composed of red LEDs (Luxeon III Emitter; Lumileds Lighting, San Jose, CA, USA) was used at a maximum wavelength of 660 nm. The LED power density at this wavelength was 22.3 mW. A reference strain of fluconazole-resistant *C. albicans* (ATCC 96901) and two clinical isolates (R10 and R15) were used. Six-week-old female Swiss mice were immunosuppressed with subcutaneous injections of prednisolone at a dose of 100 mg kg^−1^ body weight 1 day prior to inoculation with *C. albicans* (10^7^ CFU/mL) on the tongue of each mouse. On day seven, the animals were randomised to receive aPDT alone or aPDT along with LED. To perform aPDT, 70 μL of PS in saline or hydrogel was topically applied on the dorsum of the tongue. On day eight, the animals were killed, and the extracted tissues were studied by a pathologist who was blinded to the nature of the samples in order to judge the presence or absence of an inflammatory response; aPDT was very effective in the fluconazole-resistant strains of *C. albicans*.

Carmello et al. [35] used the same animal model and reported a 4.36 log10 reduction in the viability of fluconazole-susceptible *C. albicans* (ATCC 90028) using 100 mg/L PDZ and 37.5 J/cm^2^ LED light. In mice treated with PDZ-mediated aPDT for five consecutive days, aPDT was as effective as nystatin in inactivating *C. albicans* in the oral lesions of mice. The response of the strains to aPDT was not homogenous, and different *C. albicans* strains displayed differing aPDT susceptibilities. A significant reduction was observed for *C. albicans* R15 and C. albicans (ATCC 10231), while *C. albicans* R10 was not inactivated by aPDT.

#### 3.2.3. Porphyrins

Sousa et al. [36] evaluated the photodynamic effect of 5,10,-15-tris(1-methylpyridinium-4-yl)-20-(pentafluorophenyl)porphyrin tri-iodide (Tri-Py(+)-Me) and a PS formulation (FORM), based on a non-separated mixture of five cationic meso-tetraarylporphyrins, in the photoinactivation of *C. albicans* (ATCC 10231). The study was performed using blood plasma and whole blood. The porphyrin derivatives were effective in the photodynamic inactivation of *C. albicans*, promoting decreased survival of the fungus until the detection limit was reached after 30, 180, and 270 min of irradiation for FORM, Tri-Py (+)-Me, and Tetra-Py (+)-Me, respectively. It is noteworthy that FORM achieved the highest rate of photoinactivation (decrease of 0.6 log10; analysis of variance: ANOVA, *p* < 0.05) of *C. albicans* in the shortest light exposure time. The methylene blue used as the PS reference was less effective, causing a decrease of 0.8 log10 in the survival of *C.*
*albicans* after 270 min of irradiation. The study revealed the potential of the FORM and Tri-Py (+)-Me porphyrin formulations as PSs for the inactivation of *C. albicans* in blood plasma. The inactivation rates exceeded the rate found using methylene blue. Furthermore, these porphyrin PSs did not show significant negative effects on erythrocytes under isotonic conditions when haemolysis in whole blood was assessed and after the addition of treated plasma to concentrated blood cells.

Quiroga et al. [37] treated *C. albicans* suspensions with 5,10,15,20-tetrakis[4-(3 N,Ndimethylaminopropoxy)phenyl]porphyrin (TAPP) or 5,10,15,20-tetrakis[4-(3-N,N,N-trimethylaminepropoxy)phenyl]porphyrin (TAPP+4) PS and irradiated the samples for 30 min. The findings indicated the importance of the microenvironment and the effectiveness of the therapy.

The most widely used topical PS in aPDT to treat *Candida* spp. is the ester form of α-aminolevulinic acid (ALA). Cai et al. [38] described the treatment of a 64-year-old patient who presented with two lesions on his right hand and wrist. The lesions were due to *C. albicans* and had been developing for two years. The patient was treated with oral itraconazole 200 mg twice daily for 4 weeks. The treatment only slightly improved the lesions, and caused significant treatment-related adverse effects. Subsequently, the patient was treated with 10% ALA for 4 h along with PDT using 630 nm LED light (360 J/cm^2^, 100 mW/cm^2^). This procedure was repeated after 20 days. The lesions had completely disappeared two months after the last session, leaving small hypopigmented scars. The authors concluded that ALA-PDT is a good alternative to antifungal treatment. It is safe, with no major side effects, and it can be repeated as many times as necessary, since resistance does not develop.

Although aPDT is a therapy with various advantages, the most widely used PS (ALA) is water-soluble and unstable, resulting in poor bioavailability. Wang et al. [39] analysed aPDT using hexyl-aminolevulinate (HAL) as the PS. HAL is more lipophilic than ALA and can be delivered using a lipid bilayer-based vesicular delivery system (ethosomes). The authors grew *C. albicans* strain SC5314 as biofilms on YPD agar plates. Six- to eight-week-old BALB/c female mice were used to assess the pathogenicity of these strains in vivo after exposure of aPDT with 0.5% HAL delivered in the ethosomes. In strains treated with the aPDT, a significant reduction in biofilm formation was observed. Additionally, the permeability of the plasma membrane of biofilm cells was increased; microcolonies were larger, dispersed, and only rarely displayed connections between hyphae; and the cells displayed an increased susceptibility to fluconazole. In vivo, the survival rate was improved. However, ALA at 0.5%, HAL at 0.5%, and ethosomes were more damaging compared the control group. These findings likely reflected the greater formation of biofilms and mutagenesis generated as a response to stress.

The use of 5-methyl aminolevulinate (MAL) in the treatment of *Candida* spp. has been described. Aspiroz et al. [40] reported significant results in the treatment of *C. albicans* onychomycosis in a 77-year-old patient. The absorption peak was 630 nm, and the concentration of MAL was 16%. Their nails were irradiated with an Aktilite lamp (PhotoCure, Norway) at a dose of 37 J/cm^2^.

#### 3.2.4. Synthetic Dyes

Other photosensitizers used in PDT are malachite green, toluidine blue, methylene blue, and crystal violet. Malachite green is a green dye that is active against a large variety of external parasites and pathogens, including fungi and bacteria. It is a member of the triarylmethane family along with crystal violet and Victoria Blue. Absorption by red light in the visible spectrum (630 nm–700 nm) is strong. Toluidine blue is a cationic dye from the phenothiazine family. It is a derivative of aminotoluol [41]. Concerning its use as a PS, it strongly absorbs between 620 nm and 660 nm, which is within the phototherapeutic window of 600 nm–750 nm in which the penetration of light into tissues is maximum.

Wiench et al. [42] demonstrated the efficacy of aPDT against biofilms formed by *C. albicans* ATCC 10231, *C. glabrata* ATCC 15126, and *C. krusei* ATCC 14243 using a 635 nm diode laser and toluidine blue dye as the PS. In the treatment and control groups, irradiation with nine laser light wavelengths was performed using a SMARTm PRO laser diode as the light source. This source emits a continuous wave at 635 nm. An 8 mm diameter glass fibre optic head was used to avoid contact with the surface of the samples. All samples were incubated at 37 °C for 72 h. Samples were photographed, viable counts were determined, and statistical analyses were performed to determine the effect of aPDT. A statistically significant reduction was observed in *C. albicans* ATCC 10231, *C. glabrata* ATCC 15126, and *C. krusei* in the treatment and control groups for CFUs when irradiated with laser light at 400 mW and 24 J/cm^2^ for 30 s. When the appropriate laser parameters were used, toluidine blue displayed antifungal activity against the three strains. The results indicated that the effectiveness of PDT was directly dependent on the laser and the type of fungus. These in vitro findings must be validated before the PDT regimen is suitable for the clinical treatment of oral candidiasis.

The same authors also screened the titles and abstracts of related studies. Of the 32 studies that warranted in-depth scrutiny, 21 were described in a publication [43]. The materials tested in vitro were solutions of planktonic cells, planktonic cells and biofilms, cellulosic coating and seeds, and adhesive patches and seeds. Animal models included mice with infected superficial skin wounds or infected oral mucosa. The strains tested were *C. albicans* (ATCC 10231), *C. glabrata* (ATCC 15126)*, C. krusei* (ATCC14243)*, C. tropicalis* (ATCC 750) and *C.*
*parapsilosis* (ATCC 22019). The experimental groups generally included light irradiation, light irradiation in the presence of a photosensitizer, and only a photosensitizer. The applied light sources were a diode laser (maximum emission peak from 630 nm to 660 nm, with fluences between 5 J/cm^2^ and 200 J/cm^2^; 632.8 nm neodymium helium laser), lamps with a full spectrum of white light, and a fluorescent lamp. There were four different study modalities. In one, toluidine blue ortho (TBO) was the PS. In another, aPDT with TBO was compared with other PSs, which included malachite green, new methylene blue, methylene blue, rose bengal, turmeric, chlorine and erythrosine, riboflavin, pL- ce6-poly-L-lysine (e6), and S136). In the third modality, aPDT was further enhanced with Keratosis Pilaris Killer gold, chitosan, and decapeptide nanoparticles. In the fourth modality, after aPDT therapy, strains resistant to azole antifungal agents were further infused with fluconazole and posaconazole. The incubation time ranged from 30 s to 30 min. Lin et al. [44] employed chitosan for 10 min.

All the above studies demonstrated the efficacy of TBO-mediated aPDT against *Candida* spp. (reduction in the number of cells or reduction in % CFU/mL). In all yeast biofilm experiments, the partial inhibition of cell growth and reduction in mass were observed after treatment with aPDT and TBO. The weight of extracellular polymeric substance in the biofilm structure decreased by half after treatment, with a fluence of 50 J/cm^2^ and a TBO concentration of 0.1 mM. The findings have important clinical implications, as the presence of extracellular polymeric substance is important in drug resistance in biofilms, and as cells in the yeast form are more susceptible to aPDT than the filamentous form.

In studies comparing the TBO-mediated efficacy of aPDT in yeast strains, only a slightly higher effect was observed with *C. albicans* compared to other species. The addition of 0.5 µg/mL posaconazole or 0.25 µg/mL fluconazole 2 h after aPDT resulted in the eradication of *C. albicans*. The efficacy of PDT is highly dependent on several parameters. These include the light source (selected based on the PS concentration used), incubation time, concentration of the PS, and laser parameters. Concerning the PS concentration, in PDT for *Candida* spp. cells are very important. Due to the large size of the yeast cell and its more complex structure, the concentrations used to destroy the structures of the biofilms must be 100 times larger. Concerning the laser parameters, in the range of 30 J/cm^2^–40 J/cm^2^, the efficacy of the therapy can reach 80%.

TBO-mediated PDT relies on the formation of ROS, which destroy proteins, fats, and DNA, consequently damaging the cell membrane and many cell structures. The reduction in gold nanoparticles with TBO significantly reduced the survival rate of *C. albicans* in both planktonic and biofilm forms. Methylene blue, rose bengal, malachite green and riboflavin/460 nm blue light were less effective in yeast, while methylene blue N, erythrosine and chlorine (e6), and turmeric displayed a stronger effect. The use of TBO-mediated aPDT against yeast seems to be a good alternative method of monotherapy for superficial infections caused by strains that are resistant to traditional antifungal agents.

Merigo et al. [45] reported the partial effect of decapeptide on *C. albicans* biofilms (SC 5314). Curcumin had a stronger effect. Methylene blue, rose bengal, malachite green, and 460 nm riboflavin/blue light were less effective in yeast, while new methylene blue was much more effective in all cases. Wavelengths ranged from 405 nm to 650 nm.

Garcia et al. [46] compared the efficacy of aPDT on biofilms of *C. albicans* SN425 formed on two different substrates, acrylic resin and polystyrene. Two aPDT application regimens were performed: twice daily over the course of 48 h, and a single treatment after 48 h of biofilm formation. The applied PS was toluidine blue. The authors observed that the aPDT-treated samples showed a higher number of dead cells compared to the negative control with both treatment regimens. On the acrylic resin substrate, biofilms were predominantly made up of pseudohyphae. The cells were widely spread on the surface, with more space between cells compared to the biofilms formed on polystyrene plates. The findings indicated that the biofilms formed on polystyrene plates were more resistant to aPDT than biofilms formed on acrylic resin. Furthermore, applying aPDT twice daily was a better method of eradicating *C. albicans* biofilms in acrylic resin than applying aPDT after biofilms had formed. Methylene blue strongly absorbs at wavelengths greater than 620 nm, where light penetrates the tissues in an optimal way. Thus, methylene blue is an ideal compound for PDT. In PDT, methylene blue has been used to produce the ^1^O_2_ photoexcitation molecule in aqueous media. Boltes Cecatto et al. [47] reviewed 1260 articles and described the use of methylene blue concentrations between 0.0003 molar and 0.06 molar from 1 min to 5 min, while the irradiation times ranged from 8 s to 10 min.

As for the MB concentrations, 0.0003 molar to 0.06 molar was used from 1 min to 5 min, while the irradiation times ranged from 8 s to 10 min. The irradiation time ranged from 8 s to 10 min. The irradiation sources were lasers, LEDs, and lamps, with irradiances ranging from 50 mW/cm^2^ to 750 mW/cm^2^ and radiant exposures from 6 J/cm^2^ to 18 J/cm^2^.

Teichert et al. [48] explored the application of methylene blue at different concentrations (250 μg/mL, 275 μg/mL, 300 μg/mL, 350 μg/mL, 400 μg/mL, 450 μg/mL or 500 μg/mL) in the oral cavity of mice infected with *C. albicans*. After 10 min, the mice were irradiated with 664 nm diode laser light. At concentrations of 250 μg/mL to 400 μg/mL, the *C. albicans* populations were reduced. However, eradication was only possible with a concentration of 450 μg/mL to 500 μg/mL. Subsequently, the mice underwent aPDT treatment using 664 nm diode laser light (Miravant Systems, Inc., Santa Barbara, CA), 1 cm cylindrical diffuser (PDT Systems, Santa Barbara, Calif) at 275 J/cm^2^, and a fibre length at 400 mW for 687.5 s. After treatment, the mice were sacrificed, and their tongues were sampled for histopathological study. The mice in the control group were also sacrificed, and a histopathological study of the pseudomembranous lesions of the tongues was performed. There were large differences in the number of colonies in both groups. The colony size was much smaller in the treated mice. The findings indicated that methylene blue-mediated PDT is a good alternative for the treatment of mucocutaneous candidiasis.

An in vitro evaluation by Hosseini et al. [49] examined the photosensitisation effects of four distinct dyes (methylene blue, aniline blue, malachite green and crystal violet) on standard suspensions and biofilms of *C. albicans* (ATCC 5314) and *C. dubliniensis* (ATCC 6144), considering the optimum dye concentration and duration of laser irradiation. The gallium–aluminium–arsenic diode laser (Azor, Moscow, Russia) used for irradiation operated with an output power of 25 mW and a wavelength of 660 nm. Colonies of *C. albicans* and *C. dubliniensis* were irradiated for 5 min, 10 min, 20 min and 30 min. No significant difference was evident between PS alone and in combination with laser irradiation. When used alone, crystal violet led to a significant reduction in CFU compared to malachite green. Among the examined PSs, crystal violet (alone and in combination with the laser) produced the greatest reduction in CFU, although the difference was not statistically significant in comparison to the other dyes. Methylene blue dye alone did not significantly destroy *C. albicans* biofilms. Crystal violet dye had comparable effects to methylene blue and toluidine blue. Laser irradiation for 10 min resulted in the greatest reduction in CFU in comparison with the other irradiation time. Used in combination with laser irradiation, malachite green considerably reduced the CFU of the tested *Candida* species.

A system that confers antifungal resistance to *C. albicans* is the ATP-binding cassette (ABC) and the major facilitator superfamily (MFS). Oliveira-Silva et al. [50] assessed the effect of aPDT on *C. albicans* strains in the presence of glucose. This was performed since this pathogen harbours glucose sensors, which generate an intracellular signal that activates hexose transporters. This allowed the assessment of the effects of the presence of efflux pumps, such as ABC and MFS, on the efficacy of aPDT. The applied *C. albicans* strains included ATCC 10231 and YEM 12 as the wild-type controls, the MDR1 overexpressing strain YEM 13, CDR1 overexpressing strain YEM15 and YEM 14 (a wild-type relative of YEM 15). Methylene blue in solution was the PS. Irradiation was performed with an LED light emitting a wavelength 660 nm at a radiant power of 473 mW. The presence of 50 mM glucose did not have a significant effect that could generate a decrease in the sensitivity of aPDT in some of the *C. albicans* strains. The presence of MFS transporters in strains YEM12 and YEM13 improved the effect of aPDT. However, the presence of glucose in these strains generated protection from aPDT. The ABC efflux pumps present in YEM 15 protected against the lethal effect of aPDT. It is important to remember that *C. albicans* is a commensal pathogen that can adapt to various conditions, and can generate structural and metabolic changes that lead to alterations in the absorption of PSs and wavelengths of light.

Figueiredo-Godoia et al. [51] evaluated aPDT and the use of laser penetration as a treatment for candidiasis due to greater wax moths (*Galleria mellonella*). Two *C. albicans* strains were used: a clinical isolate from the oropharynx and ATCC 18804. The PS was a 10 µL volume of methylene blue. The light source was a 600 nm gallium aluminium arsenide laser beam. Fifteen *G. mellonella* larvae were used as the experimental group. The larvae were completely light and lacked dark spots in order to avoid confusion, because they could develop infection. The analyses were performed in duplicate using two control groups in each test. In one control, the applied strain of *Candida* was inoculated in the remaining progeny of the larva, after which standardised suspensions of *C. albicans* were prepared and used to inoculate larvae. In the other control, for the analysis of the survival curve, dead larvae were counted daily. Regarding the choice of the PS concentration, the output power of the laser on the sensor had to be measured. Alternatively, it could also be achieved by passing through *G. mellonella* using a calibrated power meter. For aPDT, the larvae infected with the *C. albicans* strains were injected with methylene blue dye in the last right section. They were then laser irradiated. Subsequent procedures were performed in the dark. After 6 h, the haemolymph was obtained for the determination of haemocyte counts. The results indicated that survival from candidiasis of the larvae infected with a lethal dose of fungi was prolonged by the laser treatment and aPDT.

Costa et al. [52] used rose bengal and erythrosine dyes derived from xanthene. These dyes absorb light at wavelengths of 450 nm–600 nm and 500 nm–550 nm, respectively. The authors evaluated the effects of PDT with LED irradiation of planktonic cultures and biofilms of *C. albicans*. Seven strains of the yeast isolated from the oral cavity of healthy patients along with a control strain (ATCC 18804) were used. For the sensitisation of the planktonic cultures and biofilms, concentrations of 10 µmol/L and 40 µmol/L, respectively, were used. For the planktonic cultures, 60 tests were used for each culture. After irradiation, each sample was cultured on Sabouraud Dextrose Agar plates for 48 h at 37 °C and then the CFU/mL were determined. The PDT of the biofilms was evaluated using 30 tests for each strain. Irradiation of the top surface of the planktonic suspensions and biofilm cultures was performed under aseptic conditions in the dark. For statistical analysis of the results, ANOVA and Tukey’s test were used. Scanning electron microscopy was used to analyse the *C. albicans* biofilm structure. The PDT-treated biofilms displayed reduced cell viability in their upper layers. The clinical strains also displayed a reduced number of hyphae. SEM at higher magnifications revealed cell deformation, particularly in biofilms treated with erythrosine and irradiated with LED. The results demonstrated that PDT performed with erythrosine and rose bengal with LED irradiation was effective in eliminating *C. albicans.*

Valkov et al. [53] recently tested the effects of three types of PSs, rose bengal, malachite green oxalate and methylene blue, for the elimination of *C. albicans* infections by means of aPDT. The PSs were tested at 500 µM with planktonic cultures of *C. albicans* (ATCC 90028), with prolonged illumination (1.9 ± 0.1 mW/cm^2^ for 30 min). The main objective of the study was to find an effective treatment for *C. albicans* onychomycosis. The findings revealed the superiority of rose bengal in a formulation including 5% urea and 0.5% thiourea.

Junqueira et al. [54] evaluated the in vitro sensitivity of strains of *Staphylococcus* (n = 12) and *Enterobacteriaceae* (n = 12) bacteria as well as strains of *Candida* (n = 12, including *C. albicans*, *C. tropicalis, C. parapsilosis, C. krusei* and *C. glabrata*) isolated from the oral cavity of humans, using aPDT with malachite green as the PS. Thirty-six microbial strains were included: 12 *Staphylococcus* spp., 12 Enterobacteriaceae, and 12 *Candida*. All strains were obtained from the oral cavities of patients who underwent prolonged antibiotic therapy for a minimum duration of 45 days for pulmonary tuberculosis. Subsequently, 24 tests were performed using the following treatment combinations: gallium–aluminium–arsenide laser irradiation (660 nm with a visible red-light spectrum and an output power of 35 mW) and PS; laser irradiation, as described above, and physiological solution as the light control; and PS and physiological solution as another control. A total of 864 tests were performed. Each procedure was performed in total darkness, which eventually caused an overdose in the irradiated samples.

ANOVA and Tukey’s test were used for statistical data analyses. All the laser-irradiated *Candida* strains displayed lower mean CFU counts compared to samples that were not irradiated. The findings suggested that the use of PS alone was not toxic. Additionally, *C. albicans*, *C. tropicalis*, *C. parapsilosis*, *C. krusei* and *C. glabrata* presented the same sensitivity to PDT with malachite green as the PS.

#### 3.2.5. Natural Dyes

Paz-Cristobal et al. [55] studied the susceptibility of azole-resistant *Candida* strains (AZN9635,45632Hand AMO7/0267) to PDT using two PSs (hypericin [HYP] and 1,9-dimethyl methylene blue [DMMB]) with different mechanisms of action and compared the results with those of azole-sensitive strains. ROS involved in the phototoxic effect were examined to ascertain the mechanistic differences between both PSs. The yeast cells were irradiated using LED lamps. The maximum effect was achieved after 60 min of incubation in 0.5 McFarland standard yeast suspensions. Catalase was the most active quencher for HYP in all strains, whereas superoxide dismutase, sodium azide and mannitol were less effective. The situation was different for DMMB. Sodium azide almost completely inhibited the phototoxic effect in all strains. Catalase, superoxide dismutase and mannitol were less effective. The use of ROS quenchers revealed a different pattern of ROS that contributed to the fungicidal effect of HYP and DMMB PDT. The resistance mechanisms developed by *Candida* against azole antifungals were insufficient to combat the PDT by HYP or DMMB. HYP was less effective than DMMB against strains of *C. albicans.* The findings indicated that a superior effect was realised using a low concentration of DMMB and a high dose of light.

Yang et al. [56] investigated aPDT with hypocrellin A (HA). This PS compound displayed effective antimicrobial activity when irradiation was adequate irradiation. HA is a lipid soluble perylene quinone pigment extracted from the fungus *Hypocrella bambusae*. This compound has anticancer, antimicrobial and antiviral activities. It has been used as a treatment for rheumatoid arthritis and gastric diseases, and can generate oxygen free radicals under light irradiation. The in vivo study of skin wound infections utilised 7–8-week-old female ICR mice; *C. albicans* strains SC5314, ATCC18804 and 07318; and HA (prepared as a stock solution of 1.0 mg/mL in 100% dimethylsulfoxide). Treatment with HA and 30 min of light decreased the viability of *C. albicans* SC 5314, while viability was not significantly decreased by HA treatment alone. In another experiment, mice were inoculated with 25 µL of *C. albicans* suspension. Thirty minutes later, they were treated with aPDT and HA every 24 h for seven consecutive days. The treatment reduced the wound infections and the viability of *C. albicans* in these lesions.

Although therapy with aPDT and HA showed antifungal activity, it must be considered that some fungi, such as *C. albicans*, are capable of producing melanin that can absorb light and prevent light-dependent inactivation. Hypocrellin B (HB) is a promising PS, given its higher photodynamic efficiency and lower dark toxicity toward normal cells, and is cleared more quickly from tissues. Jan et al. [57] assessed the potential of HB to mediate the photodynamic inactivation (PDI) of drug-sensitive and resistant strains of *C. albicans* and evaluated the effects of PDI on the cellular structure and surface characteristics of the yeast cells. HB did not exhibit obvious dark toxicity against *C. albicans* ATCC 10231 and azole-sensitive and resistant *C. albicans* clinical isolates. However, viability was progressively decreased after irradiation with 72 J/cm^2^ white light with increasing concentrations of HB. HB alone at a concentration of 0.1 μM produced 0.43 log_10_, 0.33 log_10_ and 0.54 log_10_ reductions in the survival of *C. albicans* ATCC 10231, the azole-sensitive clinical isolate and the azole-resistant clinical isolate. The use of the same HB concentration along with irradiation produced reductions of 2.45 log_10_, 1.73 log_10_ and 2.18 log_10_ in the same respective order. A concentration of 10 μM HB produced 4.60 log_10_, 3.54 log_10_ and 4.32 log_10_ reductions, respectively. The authors concluded that HB-mediated PDI significantly damaged the cell wall, membrane, cytoplasm and nuclei of *C. albicans*, and suggested that HB-mediated aPDT is a promising antifungal treatment.

Alshehri et al. [58] reported a study where *C. albicans* culture (ATCC 18804) had been treated with nystatin, 0.1% riboflavin in the dark, blue LED and 0.1% riboflavin for 10 min followed by photoactivation with blue LED light. Riboflavin is an essential vitamin for humans. It also has excellent photosensitive properties, with the ability to generate oxygen radicals. The authors reported that the combination of riboflavin and LED irradiation had the greatest efficacy with no side effects. This makes it very useful for the treatment of *C. albicans* in dentistry.

Dong et al. [59] used scFv phage, beginning with the selection of an scFv (JM) phage to directly recognise the MP65 mannoprotein of *C. albicans* from phage display antibody libraries. The surface-located N-terminus of the phage was modified and conjugated with pheoforbide A (PPA). The functionality of the resulting PPA-JM phage was evaluated. Confocal fluorescence microscopy images of *C. albicans* cells after incubation confirmed the binding specificities of the PPA-JM and JM phages. To verify the toxicity and selectivity of the PPA-JM phage, *C. albicans* viability was evaluated by in vitro PDT and cell viability analyses. The caspase-dependent pathway in apoptosis was induced in C. albicans by the PPA-JM phage. The implications of the induction were studied using a double iodide staining kit. Depolarization of the mitochondrial membrane potential (ΔΨm) was also analysed after PDI of PPA-JM phage-infected cells. Oxidative stress levels (ROS) were determined to reveal the alterations induced by the PPA-JM phage, which could change the pathogenicity of *C. albicans*. In these experiments, the PDI of *C. albicans* cells treated with PPA-JM phage was performed. Cell growth was inhibited, and apoptosis was subsequently observed. SEM revealed contraction and rupture of the yeast cells. When ΔΨm depolarization occurred, the intracellular levels of ROS were significantly elevated. All these events in *C. albicans* did not occur in cells infected with PPA-JM phage, in which the S phase in the cell cycle was targeted. Finally, the resulting mitochondrial dysfunction was responsible for the activation of metacaspase, which is responsible for the apoptosis of *C. albicans*. The collective findings demonstrated that the PPA-JM phage inhibits the growth and damages the surface morphology of *C. albicans*. Other detriments included apoptosis in yeast, dysfunctional mitochondria, the accumulation of ROS and the detection of S phase cells. The triggering of the apoptotic pathway in *C. albicans* by the PPA-JM phage was dependent on metacaspase. Thus, combining PS-PPA, target agents and phages could yield drugs with bifunctional antifungal therapeutic activity, expanding the horizon for the application of nano-phototherapeutics.

#### 3.2.6. Others

Ma et al. [60] studied a PS prepared from pharmacological grade aloe emodin dissolved in dimethylsulfoxide. Aloe emodin is an anthraquinone compound extracted from the leaves of the *Aloe vera* plant, as well as from the roots and rhizome of herbacious perennial plants in the genus Rheum. Antiviral, antibacterial, anticancer and anti-inflammatory activities have been attributed to aloe emodin. The authors used a standard strain of *C. albicans* (ATCC 10231) and clinical isolates that were sensitive and resistant to azoles. In the strains treated with aloe emodin solution or irradiation (96 J/cm^2^), no significant cytotoxic effects were observed. However, the combination treatment decreased the survival of the standard strain and clinical isolates. The authors also evaluated the effect of light by exposing the various *C. albicans* to aloe emodin (100 μM) in the dark for 30 min followed by irradiation at 2.4 J/cm^2^, 4.8 J/cm^2^, 14.4 J/cm^2^, or 24 J/cm^2^. The highest irradiation dose completely eradicated all *C. albicans*. Treatment with aPDT and the aloe emodin PS significantly damaged the cell wall, cytoplasm and nucleus, indicating the potential value of this approach.

Yang et al. [61] proposed a new technique to counteract intrinsic resistance. Two therapeutic modalities were used: photothermal therapy and aPDT. The combined approach was highly selectivity, non-invasive and resulted in negligible resistance. Nanotheranostics are nanoparticles that provide real-time data concerning drug biodistribution, release and targeted treatment in vivo. The amphiphilicity, good biocompatibility and lipase-sensitive biodegradability of such nanoparticles make them an excellent drug delivery platform for the construction of multimodal antifungal nanoagents. *Candida*-infected tissues have a relatively high concentration of lipase. Research has indicated the value of a lipase-triggered drug release nanoplatform comprising fluconazole nanoparticles (diketopyrrolopyrrole-polyethylene glycol (DPP-PEG)). The release of fluconazole at sites of infection can maintain a high concentration of the therapeutic drug. This approach could provide photothermogenic PSs for synergistic aPDT/photothermal therapy (PTT). The DPP–PEG–fluconazole nanoparticles comprise an antifungal platform that responds to lipase for the treatment of lesions infected by azole-resistant *C. albicans*. PEG serves as a lipase-sensitive encapsulating agent, DPP is the PS for PDT/PTT, and fluconazole is the antifungal agent. Under laser irradiation, ROS and the heat produced by the DPP–PEG–fluconazole nanoplatform render azole-resistant *C. albicans* cells more vulnerable to the antifungal agent, involving increased cell permeability. The in vivo treatment of wounds infected by *C. albicans* have demonstrated the superior antifungal activity of the nanoparticles. With the collective data, authors demonstrated the effectiveness of this therapeutic combination delivered by the nanoparticle for the treatment of azole-resistant *Candida* infections.

### 3.3. Changes in Gene Expression

It is important to note studies devoted to the study of the aPDT-mediated gene modifications of *Candida* spp. Freire et al. [62] verified the effects of PDI on *C. albicans* biofilms. The effects of aPDT on the expression of the hydrolytic enzymes aspartyl proteinase (SAP5), lipase (LIP9) and phospholipase (PLB2) were evaluated. The rational of the authors was that microorganisms do not develop resistance to PDI since there are multiple targets in fungal cells. Resistance to other antifungal agents that have a single site of action is more likely. Nine clinical isolates and the reference *C. albicans* strain ATCC 18804 were used. Methylene blue was used as the PS at a concentration of 300 µM, with an aluminium–gallium–arsenide laser light source. The biofilms were irradiated while being protected from light. RNA was extracted, quantified and purified to evaluate the expression of the SAP5, LIP9 and PLB2 genes in PDI-treated or untreated *C. albicans* by quantitative polymerase chain reaction. Gene activity was decreased by 60% for the SAP5 gene and 50% for the LIP9 and PLB2 genes following PDI. Decreases were noted in the untreated samples. The difference between the PDI and control groups was not statistically significant (*p* = 0.12, Tukey’s test). Thus, PDI with methylene blue as PS followed by low-level laser irradiation is not effective in decreasing the expression of hydrolytic enzymes of *C. albicans*.

In a subsequent study, the same researchers examined the efficacy of PDI on the expression of genes involved in *C. albicans* biofilm (ALS3, HWP1, BCR1, TEC1, CPH1 and EFG1) [63]. The effects of photosensitisation using methylene blue, low-power laser light and erythrosine sensitised with green LED were studied. Three strains of *C. albicans* were used. One was obtained from a human immunodeficiency virus-positive patient. The second was from a patient with a stomatitis lesion of the denture. The third was a control strain (ATCC 18804). The samples were seeded in HiCrome chromogenic medium and genotypically confirmed as being *C. albicans* by multiplex PCR. After identification, *C. albicans* biofilms were generated as previously described. Biofilm formation consequently involved *C. albicans* adhesion to the surface, individual colonisation and cellular organisation, secretion of extracellular polymeric substances and maturation within a three-dimensional structure, and dissemination of progeny biofilm cells. PDI of the biofilms was performed, with four laser-irradiated groups and four LED groups. Each quartet involved sensitisation with the dye and laser or LED irradiation, treatment with PS only, treatment with laser or LED irradiation only and with the photosensitizer, or no treatment. To analyse the relative changes in the qRT-PCR expression of ALS3, HWP1, BCR1, TEC1, CPH1 and EFG1 genes, the 2−ΔΔCT method was used. Statistical significance was assessed by Student’s *t* test at a level of significance of 5%. The results revealed that PDI using either laser or LED irradiation resulted in a significant temporal reduction in gene expression. The results indicated that using PDI reduced the expression of these genes in *C. albicans*, and could consequently reduce virulence in the host.

Jordão et al. [64] evaluated whether the oxidative stress produced by aPDT was responsible for altering the expression of genes related to virulence factors (adhesion and biofilm formation), as well as the expression of genes involved in the response to oxidative stress. Photodithazine (100 mg/L–200 mg/L) and curcumin (40 µM and 80 µM) were used as the PSs in the formation and treatment of biofilms of *C. albicans* ATCC 90.028. The cells were seeded in Sabouraud Dextrose Agar, and biofilms were developed over the ensuing 48 h. The PS concentrations were based on prior studies. None of the selected parameters completely eliminated *C. albicans* in treated biofilms. RNA was extracted and purified immediately after treatment. The primers used were specific for the analysis of *C. albicans* genes directly associated with adhesion and biofilm formation (ALS1 and HWP1) and oxidative stress (CAP1, CAT1 and SOD1). In the statistical analysis, analysis of variance (α = 0.05) was used, followed by Tukey’s posterior test. The results showed that aPDT using 200 mg/L photodithazine and 80 µM curcumin along with LED produced reductions in the expressions of the ALS1, HWP1, CAP1, CAT1 and SOD1 genes. Expression was also reduced by 100 mg/L photodithazine and 40 µM curcumin with LED irradiation at 50 J/cm^2^. The expression of CAP1 and SOD1 was reduced by 100 mg/L photodithazine and LED irradiation at 37.5 J/cm^2^. Another important observation was that aPDT delivered using 40 µM curcumin and irradiation at 37.5 J/cm^2^ significantly reduced the expression of the HWP1, CAP1 and SOD1 genes. The use of only LED at 37.5 J/cm^2^ down-regulated the expression of the ALS1, CAP1, CAT1 and SOD1 genes compared to the control group. The collective findings indicated the effectiveness of aPDT using photodithazine or curcumin as the PS along with LED irradiation in decreasing the expression of the evaluated genes in *C. albicans* biofilms.

Jordão et al. [65] subsequently evaluated the expression of *C. albicans* virulence genes (ALS1, HWP1, EFG1, CAT1, CAP1, SOD1, SAP1, PLB1 and LIP3), as well as genes related to the production of ergosterol (ERG1, ERG3, ERG11 and ERG25) in previously treated mice. The authors reported that aPDT eliminated the passage from yeast to hyphae, with reduced expression of the EFG1 gene. However, HWP1 expression increased in all mice treated with nystatin alone or in combination therapies. The CAP1, CAT1 and SOD1 genes are responsible for stimulating defences against free radicals and ROS; aPDT and/or nystatin decreased the expression of the CAP1 gene, which is a transcription factor in *C. albicans*. The expression of the CAT1 gene was increased in the groups treated with aPDT, nystatin and only light in comparison to the control group. The CAT1 gene is involved in the production of catalase, an important enzyme responsible for the detoxification by *C. albicans*. SOD1 expression also decreased, especially when aPDT was performed. The genes of the SOD family are responsible for defending *Candida* spp. from oxidative stress.

Alonso et al. [66] sought to determine the expressions of the ALS1, HWP1, EFG1, CAP1, CAT1, SOD1, SAP1, PLB1 and LIP3 genes, which are virulence factors in *C. albicans*. The authors used both conventional nystatin therapy and aPDT to treat *C. albicans* isolates obtained from the palates and dentures of patients from a dental school in São Paulo, Brazil. The cells were isolated at the beginning and the end of the treatment, and again 45 days later. The patients were treated with six sessions of aPDT or nystatin. In the former, 200 mg/L of photodithazine gel was applied for 20 min, followed by LED irradiation with LED at 660 nm, producing an energy density of 50 J/cm^2^ for both the prostheses and the palate. This treatment was performed three times a week for 15 days. In the nystatin group, treatment was with an oral suspension of 100,000 IU/mL nystatin delivered four times a day for 15 days. In total, 155 isolates were acquired. One patient was excluded due to insufficient RNA availability in the particular test sample. CHROM *Candida* agar was used to isolate colonies. Green colonies were suggestive of *C. albicans* and *C. dubliniensis.* Confirmation was provided by 1% agarose gel electrophoresis. Complementary DNA was synthesised in duplicate for the positive control and a single sample for the negative control. The primers used for the expression of *C. albicans* genes were specific to real-time PCR and did not cross-react with other microorganisms. A statistically significant difference in the expression of PLB1 and ACT1 was observed for both therapies, compared to the expressions of CAT1, SOD1 and LIP3 in three evaluations. Importantly, no statistically significant differences were evident in the expressions of the ALS1, HWP1, EFG1, CAP1, CAT1, SOD1, LIP3 and SAP1 genes with both therapies. Thus, the aPDT and nystatin topical treatments for Down Syndrome did not affect the expression of genes directly related to the virulence of *C. albicans.*

The main genes implicated in the genetic control of *Candida* spp. in aPDT are collated in Table 4.

## 4. Discussion

Although PDT is best known for its application in the treatment of neoplasms, there is a growing interest in aPDT, as we have described throughout this manuscript. Stájer et al. [67] have carried out an important review regarding the benefits of aPDT in the field of dentistry, as well as a detailed description of its history. Although the technique is relatively recent, its theoretical basis and application date back to the ancient Egyptians, who believed in the healing powers of the sun. Additionally, the Greeks and the civilizations of India and China described the important properties that the sun had for them. It was not until 1960, with the work performed by Lipson and Schwartz, that the use of APDT accelerated. Finally, Dougherty et al. [68] gave meaning to its applications in clinical practice.

One advantage of aPDT over different types of antifungal treatments is its specificity. The use of PS and a precise light source allows the precise triggering of reactions at a specific place and time. Many available compounds that are phototoxic properties permit the selection of an appropriate treatment strategy [21,30,31,32,33,34,35,36,37,38,39,40,41,42,43,44,45,46,47,48,49,50,51,52,53,54,55,56,57,58,59]. However, despite the many studies and efforts in the development and optimisation of effective antifungal aPDTs, in most cases, the therapeutic outcomes have been limited. There are several reasons for this. First, PSs are mostly hydrophobic molecules, which impedes their solubility. This is problematic concerning their administration and their introduction into the bloodstream. On the other hand, cell biodistribution and uptake are favoured [69].

Another drawback is that the systemic administration of PSs can lead to increased sensitivity of the skin to exposure to visible light. Patients must remain in the dark for long periods of time during treatment [70]. PSs are also subject to rapid degradation, and consequently rapid elimination from the body by cells of the reticuloendothelial system [71].

One of the crucial characteristics of the light source is the ability to generate polarised light with a specific wavelength adapted to the characteristics of the PS. There are many types of commercially available light sources. These include argon/dye lasers, metal vapour lasers, KTP:YAG/dye lasers and LED lasers. Non-laser light sources feature a light beam that can be controlled electronically or through the use of specific filters.

Adverse effects of this therapy include burning or pain. In seeking to minimise or avoid such adverse effects, new systems that use LED lamps that omit ultraviolet emissions are very promising. In addition, infrared systems do not produce light pollution and provide a narrow and highly pure band of visible light [72].

As different light sources do not radiate heat, conduction and convection mechanisms to dissipate heat are not needed. The absence of heat minimises or eliminates discomfort. Furthermore, as a low-power emission, it is not an invasive system. In this way, the undesired effects can be limited only to a transitory coloration of the area of application of the photosensitive colourant. For many authors, the light sources that we describe have disadvantages. Owing to their size, they are only suitable for external lighting. However, refinements in modern technologies include those that use optical fibres that are small in size and can be used internally in various regions of the body, such as small blood vessels or the lymphatic system [73,74].

The most studied *Candida* spp. is *C. albicans*. The data presented herein regarding genes are exclusively related to *C. albicans* [62,63,64,65,66]. Regarding the experimental models, *Candida spp.* cell lines have been a popular choice, either as strains, colonies, or biofilms. The use of mice as experimental models for oral candidiasis has been extensively documented [33,35,48]. However, the *Candida* yeast is not a habitual resident of the microbiota of mice. This greatly complicates the ability to reproduce oral candidiasis. Therefore, different models have been used to reproduce the disease in mice. In one approach, genetic modification to generate immunosuppression has been used. Another approach is the application of immunosuppressive drugs. However, as we have shown, most experimental models use both in combination. Although *C. albicans* is thought to be more resistant to PDT treatment than bacteria, which have a rigid wall, numerous studies have shown that PSs such as porphyrins, toluidine blue, methylene blue and other recently described compounds are effective in the inactivation of *C. albicans* in in vitro studies [21,30,31,32,33,34,35,36,37,38,39,40,41,42,43,44,45,46,47,48,49,50,51,52,53,54,55,56,57,58,59,60] after activation by exposure to light.

The first-generation PSs presented many drawbacks. This spurred the search for new substances with superior antimicrobial activity. Furthermore, with the introduction of nanocarriers [11,12,13,14,15,16,17,18], many obstacles posed by PS have been overcome. The refinements include increased solubility, reduced degradation, and elimination of toxicity. For example [23,24], chitosan is a natural polycationic polymer of N-acetyl-D-glucosamine and β-1,4-D-glucosamine. Natural polycations interfere with the residual negative charge of macromolecules found in the cell wall. In addition to the recognised antimicrobial action of chitosan and its action against fungi, in combination with a PS, it is able to permeabilise through the cell membrane, interact with internal cell structures and increase the antibiofilm efficacy of aPDT [75]. This binding reduces the toxicity of the PS and increases the target action of PDT [76].

Ongoing research studies are exploring new combinations of different molecules added to PSs and subsequent excitation by light. Modified fluorescent proteins, such as green fluorescent protein, can induce the production of ROS after being excited by light. The best-known protein in this group is KillerRed [77]. The intracellular location of the expressed protein can be controlled by additional genetic elements linked to the DNA sequence of the KillerRed molecule. The KillerRed protein can target the lysosomes, mitochondria and nuclei of target cells [78]. KillerRed has been used as an optogenetic tool to enable the light-mediated inactivation of various cells [79].

Another genetically encoded PS is the miniSOG flavoprotein. Chabrier-Roselló et al. [80] utilised the metabolic inhibitor antimycin A and defined genetic mutants to identify the components of the electron transport chain necessary to increase the sensitivity of PDT.

Proshkina et al. [81] demonstrated the existence of other another genetically encoded PS related to miniSOG, which appears to be applicable in PDT. Another ongoing avenue of study is theranostics. The word is derived from a combination of diagnostics and therapeutics. Theranostic agents can be classified as smart drug delivery vehicles which act by lowering pH, redox enzyme activation, elevated temperature, or magnetic fields. He et al. [82] described the use of luminogens with aggregation-induced emission. The authors reported excellent results in the PDT-mediated destruction of pathogens. Another feature of theranostics is that the treatment results can be analysed in real-time and by subsequent monitoring. Zhang et al. [83,84] described another approach termed two-photon excitation. In this approach, two independently circulating near-infrared photons reach the PS molecule simultaneously and are absorbed simultaneously. Both are equivalent to a single photon, but have half the wavelength. The need for the simultaneous absorption of both photons implies that an extremely high maximum power density is required. This, together with PDT, gives rise to high penetration power and easy penetration into tissues. He et al. [85] explained that using two-photon PDT with a high affinity for mitochondria can achieve excellent photostability and extensive generation of ROS both in vitro and in vivo.

## 5. Conclusions

PDTs were first described more than a century ago. Subsequent joint and highly multidisciplinary research involving chemists, physicists, biologists, engineers and physicians has revealed new PS molecules, and new sources of irradiation have also been developed. aPDT is an important and still-expanding field of research, reflecting the advantages of the approach over conventional antifungal treatments. However, there remain many adjustments to be made, such as optimising the physical parameters of the light sources and the establishment of clinical protocols adapted to each PS.

Most of the studies carried out with aPDT have been performed in vitro or in animal models. Almost all studies have involved *C. albicans*. More studies involving other organisms are needed in order to conclusively reveal the clinical efficacy of aPDT as a potent therapeutic strategy. It is necessary to extrapolate these data from experimental studies to clinical practice and to demonstrate their efficacy in the control of infectious processes caused by *Candida* spp. in humans.

Finally, we wish to emphasize that, in the future, nanoparticles could improve the solubility of PS, protect them from degradation, modulate their biodistribution and prolong their half-life in the blood. Thus, nanometric compounds can also improve cell uptake and achieve better target cell specificity, thus eliminating harmful side effects and systemic toxicity. Furthermore, it is necessary to develop synthesis and chemical characterization methods that allow compounds that are reproducible in terms of structure, purity and properties to be obtained. However, there are still many questions to resolve regarding the biological effects of these nanoparticles, mainly due to differences between in vitro and in vivo models.

## Figures and Tables

**Figure 1 jof-07-01025-f001:**
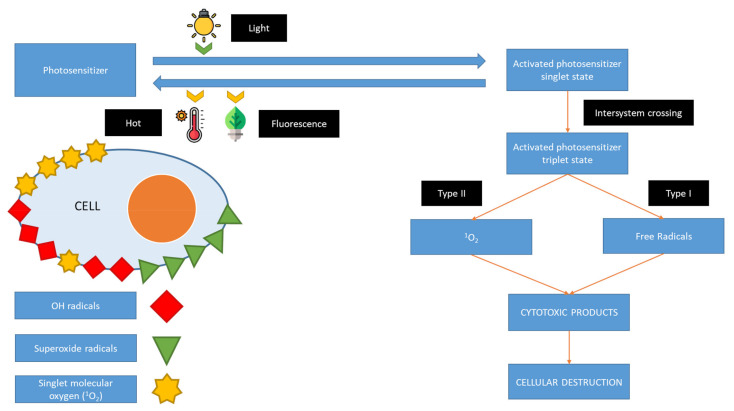
Participating elements in photodynamic reactions: Type I reactions and singlet oxygen type II reactions require the presence of molecular oxygen, superoxide, and hydroxyl radicals. ROS can affect many types of organic molecules, including nucleic acids, amino acids, and lipids.

**Figure 2 jof-07-01025-f002:**
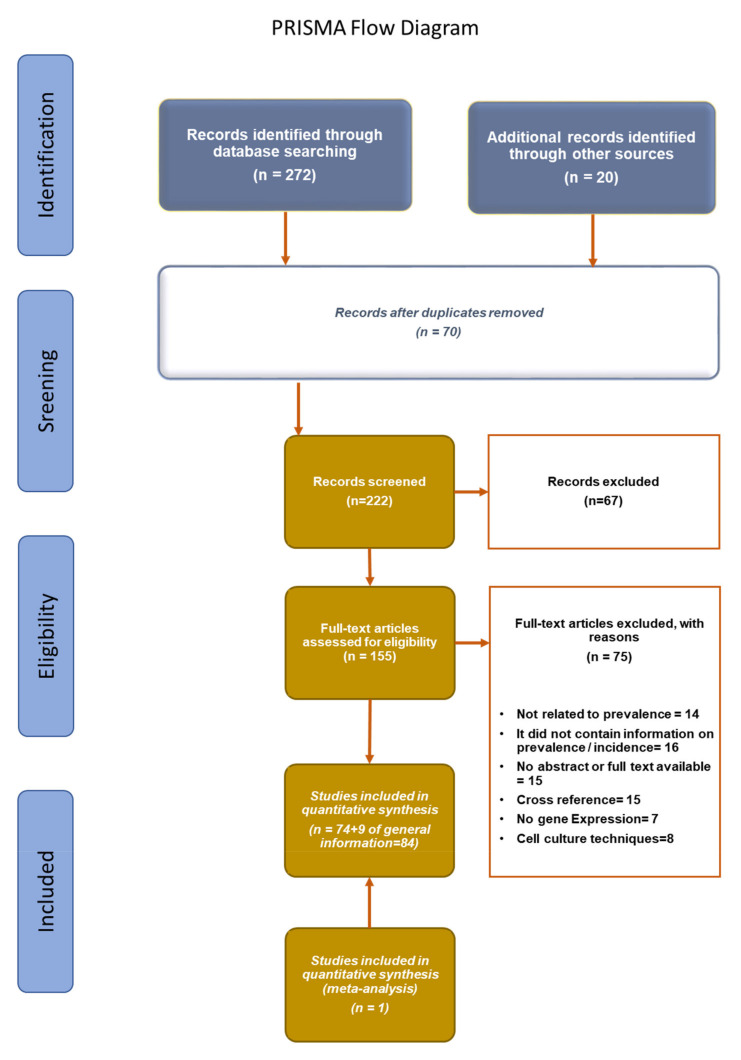
PRISMA flow diagram [10].

**Table 1 jof-07-01025-t001:** Characteristics of nanocarriers for drug delivery in antimicrobial photodynamic therapy.

Nanocarriers	Characteristics	References
Liposomes	Liposomes are spherical vesicles bounded by a membrane bilayer composed of phospholipids. The water-soluble and fat-soluble portions of the phospholipids enable the activated release of liposome contents. The main disadvantages of liposomes are their low stability and rapid elimination from the body.	Fan et al. [11] and Sercombe et al. [18]
Dendrimers	A dendrimer is a polyvalent polymer with a three-dimensional tree-like structure. They allow the transport of therapeutic agents physically entrapped within the dendritic scaffold or attached to end groups. To reduce its cytotoxicity, the use of macromolecules is necessary.	Sherje et al. [12]
Gold and silver nanoparticles	The nanoparticles consist of gold and silver particles with a typical size between 1 and 100 nm. Biodistribution and cellular absorption of the nanoparticles depend on their size and shape. There are no reliable toxicity data.	Yaqoob et al. [13]
Metal oxide nanoparticles	These are essential components of catalysts in electrochemical energy conversion and storage devices, including fuel cells, metal–air batteries, and water element separation systems. They are inorganic-based particles with a metal oxide core and coated with inorganic materials, such as silica or gold. They can also be coated by organic substances, such as phospholipids, polysaccharides, or peptides. They are non-toxic and highly biocompatible.	Zafar et al. [14]
Mesoporous silica nanoparticles	These are a mesoporous form of silica. The main characteristics of these particles are that the size of the pores and the level of porosity can be modified according to the application. They are thermally stable and biocompatible. Regarding their harmful effects, they can produce haemolysis. Prolonged retention of silica in the body can lead to carcinogenesis.	Niculescu et al. [15]
Chitosan	It is composed of fibres of marine origin, found naturally in the chitin of the shells of crustaceans. A linear polysaccharide is chemically composed of β-(1-4)-linked, deacetylated D-glucosamine and acetylated N-acetyl-D-glucosamine. Its biocompatibility and biodegradability are good. Chitosan acts synergistically with antimicrobial photodynamic therapy.	Calixto et al. [16]
Polymeric nanoparticles	They are particles of less than 1 µm in size (generally 10–500 nm). They are composed of different polymeric materials. They are stable in suspension and are readily biocompatible. As they are compounds foreign to our body, toxicity and immunogenicity can occur.	Zielińska et al. [17]

**Table 2 jof-07-01025-t002:** Examples of nanocarriers in combination with photosensitizers in antimicrobial photodynamic therapy.

Nanocarrier Type	Compound	Photosensitizer	*Candida* spp.	Effectiveness	Reference
Liposomes	CTAB-liposomes constituted with various ratios of dimyristoyl-sn-glycero-phosphatidylcholine	Chlorine e6	*C. albicans* *C. krusei* *C. tropicalis*	Improved	Yang et al. [19]
Dendrymers	Amino acid-based dendrimers and tetra- and octapeptides attached to poly(L-lysine) (PLL) dendrimers	NA	*C. albicans* *C. kefyr* *C. tropicalis*	Improved	Mlynarczyk et al. [20]
	PEGylated dendrimers	Silicon phthalocyanine Pc 4 is	*C. albicans*	Increased	Baron et al. [21]
Gold and silver nanoparticles	Monodispersed biogenic colloidal gold nanoparticles	Rose Bengal	*C. albicans*	Improved	Maliszewska et al. [22]
Chitosan	Cationic chitosan/tripolyphosphate nanoparticles	Phthalocyanine	*C. tropicalis*	Improved	Hesieh et al. [23]
	Ethylcellulose/chitosan	5,10,15,20-tetrakis(p-hydroxyphenyl)porphyrin (pTHPP)	*C. albicans*	Improved	Hasanin et al. [24]
	Carboxymethyl chitosan	1-[4-(2-aminoethyl) phenoxy] zinc (II) phthalocyanine (ZnPcN)	*C. albicans*	Increased	Tang et al. [25]
Polymeric nanoparticles	Polymeric nanocapsules	Curcumin	*C. albicans*	Variable	Sakima et al. [26]
		Zinc phthalocyanine derivatives	*C. albicans*	Improved	Evangelista et al. [27]

CTAB = cationic surfactant, cetyltrimethyl ammonium bromide. NA = not applicable or not available.

**Table 3 jof-07-01025-t003:** Overview of photosensitizer types and their characteristics.

Photosensitizer Type	Compound	Characteristics Compound Absorption Peak λ (nm) Used	Dose Range/Con-centration %	Light Dose/Power (J/cm^2^)/mW Used	*Candida* spp.	Toxicity	Outcome	References
Cyanines	Phthalocyanine	Cyanine dye; excited by red light: λ = 660 nm and 850 nm; penetrability 1–3 h after intravenous administration	0.14 mg/kg intravenously	12.6–94.5 J/cm^2^	Colonies *C. albicans* ATCC 10231	Minimal skin photosensitivity	To increase efficiency, nanocarriers are needed	Malisze-wska et al. [30]
	Indomethacin green	λ = 606 and 808 nm	0.1 mL	10 J/cm^2^	*C. albicans* suspensions ATCC 10231	NA	Effective	Azizi et al. [31]
	Silicon phthalocyanine Pc 4	λ = 670–675	5 µM of Pc4 encapsulated in PEGylated PAMAM nanoparticles	10 J/cm^2^	*C. albicans* strains SC 5314	NA	Effective	Hutnick et al. [21]
Chlorins	Chlorin e6 plus graphene	λ = 632 nm during 15 min	2 mL	150 mW	Cells of *C. albicans* ATCC 90028	NA	Effective	Acosta et al. [32]
	Photoditazine^®^ (PDZ) consists of a N-methyl-D-glucosamine	λ = 660 nm during 2–8 min	200 mg/L	50 J/cm^2^ 690 mW during 19 min	Tongues of mice with oral candidiasis: *C. albicans* ATCC 96901	NA	Efficacy of therapies evaluated by microbiological, macroscopic, histopathological and confocal scanning laser microscopy	Hidalgo et al. [33]
		λ = 650–680 nm	100 mg/L	37.2 J/cm^2^ during 14 min	Tongues of mice with oral candidiasis: *C. albicans* ATCC 10231	NA	Effective, but in vivo studies needed	Alves et al. [34]
		λ = 660 nm	100 mg/L	37.5 J/cm^2^ 22.3 mW	Tongues of mice with oral candidiasis: *C. albicans* ATCC 90028	Safe	Effective	Carmello et al. [35]
Porphyrins	Hematoporphyrin and porfimer sodium	Tetrapyrrole structure; excited by red light: λ = 660- nm-635 nm; penetrability time a 20 min	0.5–5 mg/kg Intravenous-ly	75–250 J/cm^2^	*C. albicans* cells in suspension ATCC 10231	Great photosensitivity for 6–10 weeks after treatment	Effective	Sousa et al. [36]
	(ALA 5,10,15,20-tetrakis[4-(3 N,Ndimethylaminopropoxy)phenyl]porphyrin (TAPP) and 5,10,15,20-tetrakis[4-(3-N,N,N-trimethylaminepropoxy)phenyl] porphyrin	λ = 350–800 nm during 30 min	5 μM	90 mW/cm^2^	*C. albicans* strain PC31	NA	Effective	Quiroga et al. [37]
		Tetrapyrrole structure 630–760 nm ALA and MAL are protoporphyrin IX precursors and, therefore, inactive drugs	10–60 mg/kg orally 10–30% topically 3–6% Intravesicularly	50–150 J/cm^2^	*C. albicans* (in vivo) NA	Neurotoxicity	Effective	Cai et al. [38]
	MAL) Hexyl-aminolevulinate and ethosomes	λ = 633 ± 10 nm	100 µL of 0.5% hexyl-aminolevulinate and ethosomes	60 mW/cm^2^, distance 10 cm	*C. albicans* strain SC 5314	NA	Effective	Wang et al. [39]
	5-Methyl-aminolevulinate	λ= 630 nm	16–20%	37 J/cm^2^	*C. albicans* (in vivo) NA	NA	Effective	Aspiroz et al. [40]
Synthetic Dyes	Toluidine blue	λ = 635 nm	NA	400 mW 24 J/cm^2^ 300 mW 18 J/cm^2^ 20,000 mW 22 J/cm^2^	*Strains of C. albicans*, ATCC 10231 *C. glabrata* *ATCC* *15126* and *C. krusei* *ATCC* *14243*	NA	Variable, depends on yeast type	Keten et al. [41] and Wiench et al. [42]
	Toluidine ortho blue	λ = 630–660 nm	0.1 μM	50 J/cm^2^	*Strains of C. albicans*, ATCC 10231 *C. glabrata* *ATCC* *15126* and *C. krusei* *ATCC* *14243**, C. tropicalis* ATCC 750 and *C.* *parapsilosis* ATCC 22019.	NA	Efficacy reaches 80%	Wiench et al. [43] and Lin et al. [44]
	Toluidine blue	λ = 405–650 nm	20 μg/mL	10 J/cm^2^	*C. albicans* SC5314	NA	Effective	Merigo et al. [45]
		λ = 635 nm	44 μM	175.2 J/cm^2^, 2 min	*C. albicans* biofilms SN425	NA	Variable	Garcia et al. [46]
	Methylene blue	λ = 635 nm	0.0003 to 0.06 molar	6- 18 J/cm^2^ 50–750 Mw, from 8 s to 10 min	*C. albicans* in vivo NA	NA	Variable	Boltes Cecatto et al. [47]
		λ = 664 nm	25–500 μg/mL	275 J/cm2 and 400 mW during 687.5 s	*Candida albicans* NA	NA	Effective	Teichert et al. [48]
		λ = 660 nm	0.01–0.001 mg/mL	76.8 J/cm^2^ and 25 mW, 5–30 min	*C. albicans* ATCC 5314 and *C. dubliniensis* ATCC 6144	NA	Partially effective	Hosseini et al. [49]
	Methylene blue and glucose	ʎ = 660 nm at a radiant power of 473 mW and 50 mM glucose	0 mM glucose 100 μM MB	10, 30 and 60 J/cm^2^ 473 mW	Cells and biofilm *C. albicans* ATCC 10231	NA	Partial effect	Oliveira-Silva et al. [50]
	Methylene blue and *G. mellonella*	λ = 660 nm	10 µL	10–15 J/cm^2^	*C. albicans* strain ATCC 18804	NA	Effective	Figueiredo-Godoi et al. [51]
	Rose Bengal	Rose Bengal excited by green light at λ = 450–600 nm	Cultures 10 µmol/L; biofilms 40 µmol/L	36 J/cm^2^ for 180 s	Planktonic cultures and biofilms of *C. albicans* ATCC 18804	NA	Reduction in number of hyphae	Costa et al. [52]
	Crystal violet	N-tetra-penta-hexametil p-rosanilinas; λ =660 nm	1 g/10 mL	76.8 J/cm^2^, 25 mW for 5–30 min	*C. albicans* ATCC 5314 and *C. dubliniensis* ATCC 6144	NA	Effective	Hosseine et al. [49]
	Malachite green	400–700 nm	50–500 µM	1.9 ± 0.1 mW/cm^2^ for 30 min	Planktonic cultures of *C. albicans* ATCC 90028	NA	Effective	Valkov et al. [53]
	Malachite green	λ = 660 nm	0.1 mL	26 J/cm 35 mW during 4.45 min	Microbial suspensions containing 10^6^ cells/mL *C. albicans* *C. tropicalis*, *C. parapsilosis*, *C. krusei* and *C. glabrata* In vivo NA	NA	Low cost, highly efficient, and short application time	Junqueira et al. [54]
Natural dyes	Hyperecin	λ = 570 nm	0.62 μmol/L	18–37 J/cm^2^	*C. albicans* strains AZN 9635,45632Hand AMO7/0267	NA	Effective	Paz-Cristobal et al. [55]
	Hipocrellin A and B	Peak at λ = 600–900 nm; λ = 470 nm	1 mg/mL to 10 μM	72 J/cm^2^ 30 min	*C. albicans* strains SC 53114, ATCC 18804 and 07318	NA	Effective	Yang et al. [56] and Jan et al. [57]
	Riboflavin	Λ = 360 and 440 nm	0,1%	blue LED light (light dose)	Culture of *C. albicans* ATCC 18804	NA	Effective	Alshehri et al. [58]
Chlorophyll	Pheophorbide A conjugated with JM-phage by EDC/NHS crosslinking	Tetrapyrrole structure λ = 670 nm	5 μM	50 mW/cm^2^ for 10 min	*C albicans* cells ATCC 10231	NA	Effective	Dong et al. [59]
Others	Aloe-emodin (1,8-dihydrox 3-(hydroxymethyl)-9,10-anthracenedione)	λ = 400–780 nm	0.1–100 μM for 30 s to 5 min	4.8 J/cm^2^ during 1 min	*C. albicans* cell suspension ATCC 10231	NA	Effective	Ma et al. [60]

PS = photosensitizer; ALA = 5-aminolevulinic acid; MAL = 5-methyl aminolevulinate; s = second; min = minute; g = gram; mg = milligram; μM = micromole; λ = wavelength; nm = nanometres; J/cm^2^ = Joules per square centimetre; NA = not applicable or not available.

**Table 4 jof-07-01025-t004:** Characteristics of C. albicans genes related to virulence and involved in antimicrobial photodynamic therapy.

*Candida* spp.	Photosensitizer	Genes	Results	Origin	References
*C. albicans* ATCC 18804	Methylene blue	SAP5, LIP9 and PLB2	60% SAP5/50% LIP9 and PLB2	Biofilms	Freire et al. [62]
*C. albicans* ATCC 18804	Methylene blue	ALS3, HWP1, BCR1, TEC1, CPH1 y EFG1	ALS3, HWP1, BCR1, TEC1, CPH1 y EFG1	Strains *C. albicans* from patients with a control strain (ATCC 18804)	Freire et al. [63]
*C. albicans* ATCC 90028	Photodithazine (PDZ and Curcumin	ALS1, HWP1, CAP1, CAT1, SOD1	ALS1, HWP1, CAP1, CAT1, SOD1	Biofilms	Jordão et al. [64]
*C. albicans* ATCC 96901	Photodithazine^®^ (PDZ)	ALS1, HWP1, EFG1, CAT1, CAP1, SOD1, SAP1, PLB1 and LIP3	Special decrease in SOD1	Colonies from tongues of mice	Jordão et al. [65]
*C. albicans* ATCC 90028	Photodithazine^®^ (PDZ) plus Nystatine	ALS1, HWP1, EFG1, CAP1, CAT1, SOD1, SAP1, PLB1 y LIP3	PLB1 y ACT1	Dental patient samples	Alonso et al. [66]

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
