# Peer review of "New Applications of Photodynamic Therapy in the Management of Candidiasis"

_jof, 2021, doi:10.3390/jof7121025_

Round 1
Reviewer 1 Report
Throughout the document it is evident that when referring to different species of Candida; "Candida spp", the authors do it inadequately. The correct form is: Candida spp., only the genus is written in italics. The (spp.,) should be written without italics and accompanied by a period and then a comma as it is shown.
Line 27: the word “recognized” is misspelled. I suggest that the manuscript be submitted to English revision, as the service provided by JoF.
Lines 87 - 91: All species names must be italicized. Please check throughout the manuscript, including references.
Lines 97, 300, 475, 501, 554, 615, 654, 668, 767, 769: the genus "Candida" must be written in italics, please standardize throughout the entire manuscript.
Lines 575, 577: Strain CS 5314 is misspelled "C5314". Please correct.
Lines 134, 186, 482, 590: please remove the “f” and comma from “C. albicans fATCC 10,231” the correct way to write this reference strain is: C. albicans ATCC 10231
Line 266: when you are talking about XTT, the context of “ATCC” is not clear. Rephrase the sentence please.
Line 287: indicate the quantity of prednisolone expressed in mg / kg that was used in the animal model with Swiss mice.
Line 292: please Correct “fluconazole-resistant” for fluconazole-resistant
Line 363: please Correct “ATCC 14,243” for C. krusei ATCC 14243
Line 676: please Correct “ATCC 18,804” for C. albicans ATCC 18804
Line 712: please Correct “ATCC 90,028” for C. albicans ATCC 90028
Lines 382, 383, 437, 449: Please indicate specifically the strains of C. albicans, C. glabrata, C. krusei, C. tropicalis, and C. parapsilosis that were analyzed or clarify if they are clinical isolates or even unidentified strains that are causing a recent infection process.
Línea 632: there is a period left.
Lines 872 and 873: Reconsider the final sentence of the conclusion where the authors indicate that it is necessary to explore more data from experimental studies to demonstrate the efficacy in the eradication of Candida spp., in humans. The objective of these investigations throughout the world is not to eradicate Candida, as it is important to note that this yeast is part of the human being natural microbiota, but then to control the infectious processes caused by this microorganism.
Tables 2 and 3: please review the tables design so that the words are not cut off, if it is necessary you could reduce a little bit the font size.
I suggest that the information in table 3 could be cited in the text and the table shown as complementary information.
Table 4: in the left column, please list the C. albicans strains used
Author Response
First of all, I would like to thank the reviewer 1 for the work they have done in correcting our manuscript, which has been very helpful. Below we have answered the questions and addressed the corrections suggested to us:
Reviewer: 1
Comments to the Author
NUMBER 1
(x) I would not like to sign my review report
( ) I would like to sign my review report
English language and style
( ) Extensive editing of English language and style required
(x) Moderate English changes required
( ) English language and style are fine/minor spell check required
( ) I don't feel qualified to judge about the English language and style
|
Is the work a significant contribution to the field? |
|
|
Is the work well organized and comprehensively described? |
|
|
Is the work scientifically sound and not misleading? |
|
|
Are there appropriate and adequate references to related and previous work? |
|
|
Is the English used correct and readable? |
Comments and Suggestions for Authors
Throughout the document it is evident that when referring to different species of Candida; "Candida spp", the authors do it inadequately. The correct form is: Candida spp., only the genus is written in italics. The (spp.,) should be written without italics and accompanied by a period and then a comma as it is shown.
Line 27: the word “recognized” is misspelled. I suggest that the
Line 27: the word “recognized” is misspelled. I suggest that the manuscript be submitted to English revision, as the service provided by JoF.
Lines 87 - 91: All species names must be italicized. Please check throughout the manuscript, including references.
Lines 97, 300, 475, 501, 554, 615, 654, 668, 767, 769: the genus "Candida" must be written in italics, please standardize throughout the entire manuscript.
Lines 575, 577: Strain CS 5314 is misspelled "C5314". Please correct.
Lines 134, 186, 482, 590: please remove the “f” and comma from “C. albicans fATCC 10,231” the correct way to write this reference strain is: C. albicans ATCC 10231
Line 266: when you are talking about XTT, the context of “ATCC” is not clear. Rephrase the sentence please.
Line 287: indicate the quantity of prednisolone expressed in mg / kg that was used in the animal model with Swiss mice.
Line 292: please Correct “fluconazole-resistant” for fluconazole-resistant
Line 363: please Correct “ATCC 14,243” for C. krusei ATCC 14243
Line 676: please Correct “ATCC 18,804” for C. albicans ATCC 18804
Line 712: please Correct “ATCC 90,028” for C. albicans ATCC 90028
Lines 382, 383, 437, 449: Please indicate specifically the strains of C. albicans, C. glabrata, C. krusei, C. tropicalis, and C. parapsilosis that were analyzed or clarify if they are clinical isolates or even unidentified strains that are causing a recent infection process.
Línea 632: there is a period left.
Lines 872 and 873: Reconsider the final sentence of the conclusion where the authors indicate that it is necessary to explore more data from experimental studies to demonstrate the efficacy in the eradication of Candida spp., in humans. The objective of these investigations throughout the world is not to eradicate Candida, as it is important to note that this yeast is part of the human being natural microbiota, but then to control the infectious processes caused by this microorganism.
Tables 2 and 3: please review the tables design so that the words are not cut off, if it is necessary you could reduce a little bit the font size.
I suggest that the information in table 3 could be cited in the text and the table shown as complementary information.
Table 4: in the left column, please list the C. albicans strains used
Authors' reply
We did all the suggestions and changes proposed by the reviewer. The manuscript is sent with the corrections marked in yellow

Reviewer 2 Report
An interesting review article describing the advancement of photodynamic therapy in the management of Candidiasis. This review is well written and does a thorough job describing the current landscape of this potential form of anti-fungal therapy, however there are a few items that I feel should be addressed prior to publication.
Tables presented in the review should be of proper formatting size to fit the document, text in the tables are larger in font size as compared to the manuscript itself and in many cases the text within the tables was being truncated and wrapped down to the next line because it was too large to fit within the boundaries of the table. This is something that can be fixed with proper font sizes.
The manuscript itself, while comprehensive of the subject matter, does not flow very well for the reader. Each paragraph in the "results" section exists to purely summarize one of the references in this manuscript. As a review article I don't feel this is the best way to present the collective data and perhaps the review would flow better if the authors created specific categories within the manuscript and wrote accordingly, ie a section discussing Cyanines, Chlorins, etcetera.
Author Response
First of all, I would like to thank the reviewer 2 for the work they have done in correcting our manuscript, which has been very helpful. Below we have answered the questions and addressed the corrections suggested to us:
Reviewer: 2
Comments to the Author
(x) I would not like to sign my review report
( ) I would like to sign my review report
English language and style
( ) Extensive editing of English language and style required
( ) Moderate English changes required
(x) English language and style are fine/minor spell check required
( ) I don't feel qualified to judge about the English language and style
|
Is the work a significant contribution to the field? |
|
|
Is the work well organized and comprehensively described? |
|
|
Is the work scientifically sound and not misleading? |
|
|
Are there appropriate and adequate references to related and previous work? |
|
|
Is the English used correct and readable? |
Comments and Suggestions for Authors
An interesting review article describing the advancement of photodynamic therapy in the management of Candidiasis. This review is well written and does a thorough job describing the current landscape of this potential form of anti-fungal therapy, however there are a few items that I feel should be addressed prior to publication.
Tables presented in the review should be of proper formatting size to fit the document, text in the tables are larger in font size as compared to the manuscript itself and in many cases the text within the tables was being truncated and wrapped down to the next line because it was too large to fit within the boundaries of the table. This is something that can be fixed with proper font sizes.
The manuscript itself, while comprehensive of the subject matter, does not flow very well for the reader. Each paragraph in the "results" section exists to purely summarize one of the references in this manuscript. As a review article I don't feel this is the best way to present the collective data and perhaps the review would flow better if the authors created specific categories within the manuscript and wrote accordingly, ie a section discussing Cyanines, Chlorins, etcetera.
Authors' reply
We did all the suggestions and changes proposed by the reviewer. The manuscript is sent with the corrections marked in yellow.

Reviewer 3 Report
Dear Authors,
The manuscript ID: jof-1445720 entitled „New applications of photodynamic therapy in the management of Candidiasis” written by Carmen Rodríguez-Cerdeira, Erick Martínez Herrera, Gabriella Fabbrocini, Beatriz Sanchez-Blanco, Adriana López-Barcenas, May EL-Samahy, Eder R. Juárez-Durán and José Luís González-Cespón is very interesting.
Over the past years, the occurrence of both superficial and systemic, life-threatening fungal infections caused by Candida spp. has increased dramatically. Currently, the list of the commertially available antifungal agents, used for the treatment of candidiasis, is limited to three major classes: polyenes, azoles, and echinocandins. Moreover, the these infections are difficult to treat due to resistance to many antimycotics, especially azoles. Their treatment is often ineffective, so there is a need to search for new antifungal therapies. According to me, the subject of the work is extremely relevant and topical. Photodynamic therapy has been recognized as an alternative method of pathogen inactivation, is important in the fight against yeast, and its new applications could be widely used in the treatment of candidiasis.
The whole manuscript (Introduction, Materials and Methods, Results, Discussion, Conclusions) is properly organized. Introduction contains general data on candidiasis and photodynamic therapy. Results are widely documented. Based on the results, adequate discussion and conclusions were drawn. This comprehensive, systematic review was undertaken to consolidate the new applications of photodynamic therapy in the management of candidiasis. It is a well written and original review.
I have only some suggestions in order to improve paper, which are the following:
- In the whole text: spp. – spp. please write without italics
- Table 1, Table 2, Table 3 – please standardize the tables, font, remove highlights
- please correct very minor mistakes:
Line 3: Candidiasis – candidiasis
Lines 132, 475, 501, 554, 654, 668, 767: Candida – write in italics, please
Line 134: C. albicans fATCC 10,231 – C. albicans ATCC 10231
Lines 183-185: please standardize the font
Line 186: C. albicans ATCC 10,231 – C. albicans ATCC 10231
Line 300: C. albicans – write in italics, please
Line 363: C. krusei ATCC 14,243 – C. krusei ATCC 14243
Line 364: PS . – PS.
Lines 519, 713: Sabouraud Dextrose Agar – sabouraud dextrose agar
Lines 537, 540: Enterobacteriaceae – write in italics, please
Line 569: aPDT) – aPDT
Line 575: ATCC18804 – ATCC 18804
Line 590: ATCC 10,231 – ATCC 10231
Lines 614-615: C. albicans – write in italics, please
Lines 676-677: ATCC 18,804 – ATCC 18804
Line 818: [ 30-60] – [30-60]
- Please unify the literature
According to me, this manuscript is valuable and may be accepted for the publication in “Journal of Fungi”.
With highest regards,
Author Response
First of all, I would like to thank the reviewer 3 for the work they have done in correcting our manuscript, which has been very helpful. Below we have answered the questions and addressed the corrections suggested to us:
Reviewer: 3
Comments to the Author
Open Review
(x) I would not like to sign my review report
( ) I would like to sign my review report
English language and style
( ) Extensive editing of English language and style required
( ) Moderate English changes required
( ) English language and style are fine/minor spell check required
(x) I don't feel qualified to judge about the English language and style
Is the work a significant contribution to the field?
Is the work well organized and comprehensively described?
Is the work scientifically sound and not misleading?
Are there appropriate and adequate references to related and previous work?
Is the English used correct and readable?
Comments and Suggestions for Authors
Dear Authors,
The manuscript ID: jof-1445720 entitled „New applications of photodynamic therapy in the management of Candidiasis” written by Carmen Rodríguez-Cerdeira, Erick Martínez Herrera, Gabriella Fabbrocini, Beatriz Sanchez-Blanco, Adriana López-Barcenas, May EL-Samahy, Eder R. Juárez-Durán and José Luís González-Cespón is very interesting.
Over the past years, the occurrence of both superficial and systemic, life-threatening fungal infections caused by Candida spp. has increased dramatically. Currently, the list of the commertially available antifungal agents, used for the treatment of candidiasis, is limited to three major classes: polyenes, azoles, and echinocandins. Moreover, the these infections are difficult to treat due to resistance to many antimycotics, especially azoles. Their treatment is often ineffective, so there is a need to search for new antifungal therapies. According to me, the subject of the work is extremely relevant and topical. Photodynamic therapy has been recognized as an alternative method of pathogen inactivation, is important in the fight against yeast, and its new applications could be widely used in the treatment of candidiasis.
The whole manuscript (Introduction, Materials and Methods, Results, Discussion, Conclusions) is properly organized. Introduction contains general data on candidiasis and photodynamic therapy. Results are widely documented. Based on the results, adequate discussion and conclusions were drawn. This comprehensive, systematic review was undertak
adequate discussion and conclusions were drawn. This comprehensive, systematic review was undertaken to consolidate the new applications of photodynamic therapy in the management of candidiasis. It is a well written and original review.
I have only some suggestions in order to improve paper, which are the following:
- In the whole text: spp. – spp. please write without italics
- Table 1, Table 2, Table 3 – please standardize the tables, font, remove highlights
- please correct very minor mistakes:
Line 3: Candidiasis – candidiasis
Lines 132, 475, 501, 554, 654, 668, 767: Candida – write in italics, please
Line 134: C. albicans fATCC 10,231 – C. albicans ATCC 10231
Lines 183-185: please standardize the font
Line 186: C. albicans ATCC 10,231 – C. albicans ATCC 10231
Line 300: C. albicans – write in italics, please
Line 363: C. krusei ATCC 14,243 – C. krusei ATCC 14243
Line 134: C. albicans fATCC 10,231 – C. albicans ATCC 10231
Lines 183-185: please standardize the font
Line 186: C. albicans ATCC 10,231 – C. albicans ATCC 10231
Line 300: C. albicans – write in italics, please
Line 363: C. krusei ATCC 14,243 – C. krusei ATCC 14243
Line 364: PS . – PS.
Lines 519, 713: Sabouraud Dextrose Agar – sabouraud dextrose agar
Lines 537, 540: Enterobacteriaceae – write in italics, please
Line 569: aPDT) – aPDT
Line 575: ATCC18804 – ATCC 18804
Line 590: ATCC 10,231 – ATCC 10231
Lines 614-615: C. albicans – write in italics, please
Lines 676-677: ATCC 18,804 – ATCC 18804
Line 818: [ 30-60] – [30-60]
- Please unify the literature
According to me, this manuscript is valuable and may be accepted for the publication in “Journal of Fungi”.
With highest regards,
Authors' reply
We did all the suggestions and changes proposed by the reviewer. The manuscript is sent with the corrections marked in yellow.

Reviewer 4 Report
General:
please adhere to the International guidelines and taxonomic principles for writing bacterial names
Gram-positive, Gram-negative
please describe all abbreviations on their first mention
Abstract:
L24: …aetiological agents
L24: …are Candida spp.
L25: …which may be fatal
L27: what do you mean by pathogen inactivation? please rephrase!
Introduction:
In the first section, please elaborate in a more detailed extent on the most relevant virulence factors of Candida spp. from the standpoint of oral infections, in addition to the underlying factors facilitating the development of invasive infections!
L40: may lead to an infection, facilitated by…
L41: Consequences of these pathologies may range from…
L44: and may result in
L54: please elaborate more on the therapeutic difficulties of candidasis to put the significance of aPDT in more context
L77: I would suggest the brief classification of PSs, based on their chemical structure
Methods:
this syst. rev. was undertaken and reported via PRISMA guidelines, which is in line with current standards
I would suggest that the reasons for exclusion (n=97) in the relevant box of the flow-chart be elaborated more, as they are not understandable and quite ambiguous in its current form.
Results
Table 1.: adequate, relevant data is presented
L134: ATCC 10231
L145: may be eliminated and the…
Table 2.: in the column „Candida spp.” you should include the strain codes (e.g. ATCC) from the respective articles
L170: please double-check the names of the aPDTs
Table 3.: in the column „Candida spp.” you should include the strain codes (e.g. ATCC) from the respective articles
L186: ATCC 10231
L217: encapsulation
L262: ATCC 90028
L185-L395: I would suggest the re-arrangement of these articles in a more systematic and logical fashion. In addition, I would suggest breaking up huge blocks of text.
L363: ATCC 14243
L442-443: between 0.0003 to 0.06 molar. please correct
L482: ATCC 10231
L537: Enterobacteriaceae in italics
L590: ATCC 10231
3.3. Changes in gene expression
Table 4.: in the column „Candida spp.” you should include the strain codes (e.g. ATCC) from the respective articles
Discussion:
Please provide a brief overview on the history of aPDT and the current concepts.
If appropriate, please consider including the following reference:
https://pubmed.ncbi.nlm.nih.gov/32392793/
Conclusions:
Please rewrite and make the conclusions a bit more comprehensive.
Author Response
First of all, I would like to thank the reviewer 4 for the work they have done in correcting our manuscript, which has been very helpful. Below we have answered the questions and addressed the corrections suggested to us:
Reviewer: 4
Comments to the Author
(x) I would not like to sign my review report
( ) I would like to sign my review report
English language and style
( ) Extensive editing of English language and style required
( ) Moderate English changes required
(x) English language and style are fine/minor spell check required
( ) I don't feel qualified to judge about the English language and style
|
Is the work a significant contribution to the field? |
|
|
Is the work well organized and comprehensively described? |
|
|
Is the work scientifically sound and not misleading? |
|
|
Are there appropriate and adequate references to related and previous work? |
|
|
Is the English used correct and readable? |
Comments and Suggestions for Authors
General:
please adhere to the International guidelines and taxonomic principles for writing bacterial names
Gram-positive, Gram-negative
please describe all abbreviations on their first mention
Abstract:
L24: …aetiological agents
L24: …are Candida spp.
L25: …which may be fatal
L27: what do you mean by pathogen inactivation? please rephrase!
Introduction:
In the first section, please elaborate in a more detailed extent on the most relevant virulence factors of Candida spp. from the standpoint of oral infections, in addition to the underlying factors facilitating the development of invasive infections!
L40: may lead to an infection, facilitated by…
L41: Consequences of these pathologies may range from…
L44: and may result in
L54: please elaborate more on the therapeutic difficulties of candidasis to put the significance of aPDT in more context
L77: I would suggest the brief classification of PSs, based on their chemical structure
Methods:
this syst. rev. was undertaken and reported via PRISMA guidelines, which is in line with current standards
I would suggest that the reasons for exclusion (n=97) in the relevant box of the flow-chart be elaborated more, as they are not understandable and quite ambiguous in its current form.
Results
Table 1.: adequate, relevant data is presented
L134: ATCC 10231
L145: may be eliminated and the…
Table 2.: in the column „Candida spp.” you should include the strain codes (e.g. ATCC) from the respective articles
L170: please double-check the names of the aPDTs
Table 3.: in the column „Candida spp.” you should include the strain codes (e.g. ATCC) from the respective articles
L186: ATCC 10231
L217: encapsulation
L262: ATCC 90028
L185-L395: I would suggest the re-arrangement of these articles in a more systematic and logical fashion. In addition, I would suggest breaking up huge blocks of text.
L363: ATCC 14243
L442-443: between 0.0003 to 0.06 molar. please correct
L482: ATCC 10231
L537: Enterobacteriaceae in italics
L590: ATCC 10231
3.3. Changes in gene expression
Table 4.: in the column „Candida spp.” you should include the strain codes (e.g. ATCC) from the respective articles
Discussion:
Please provide a brief overview on the history of aPDT and the current concepts.
If appropriate, please consider including the following reference:
https://pubmed.ncbi.nlm.nih.gov/32392793/
Conclusions:
Please rewrite and make the conclusions a bit more comprehensive.
Authors' reply
We did all the suggestions and changes proposed by the reviewer. The manuscript is sent with the corrections marked in yellow.

Round 2
Reviewer 1 Report
I believe the authors made the suggested changes. However, when writing the name of the species, they must remember that after the spp it must be followed by a period and comma (Candida spp., Xxxx). At the same time, it would be important analyze the possibility of placing table 3 in supplementary material. If this is not possible, it would be important to better redistribute column 6 (Candida spp.) So that the words do not are cut off.
Author Response
First of all, I would like to thank the reviewer 1 for the work they have done in correcting our manuscript, which has been very helpful. Below we have answered the questions and addressed the corrections suggested to us:
Reviewer: 1
Comments to the Author
Open Review
( ) I would not like to sign my review report
(x) I would like to sign my review report
English language and style
( ) Extensive editing of English language and style required
(x) Moderate English changes required
( ) English language and style are fine/minor spell check required
( ) I don't feel qualified to judge about the English language and style
|
Is the work a significant contribution to the field? |
|
|
Is the work well organized and comprehensively described? |
|
|
Is the work scientifically sound and not misleading? |
|
|
Are there appropriate and adequate references to related and previous work? |
|
|
Is the English used correct and readable? |
Comments and Suggestions for Authors
I believe the authors made the suggested changes. However, when writing the name of the species, they must remember that after the spp it must be followed by a period and comma (Candida spp., Xxxx).
Authors' reply
Dear reviewer, we have verified that both the editor of the Journal of Fungi, as well as the Guest Editors of this issue, use and agree to put Candida spp. We send you bibliography about it. Hope you understand us.
Thank you
- Camp I, Spettel K, Willinger B. Molecular Methods for the Diagnosis of Invasive Candidiasis. J Fungi (Basel). 2020 Jul 6;6(3):101. doi: 10.3390/jof6030101. PMID: 32640656; PMCID: PMC7558065.
- Mareković I, Pleško S, Rezo Vranješ V, Herljević Z, Kuliš T, Jandrlić M. Epidemiology of Candidemia: Three-Year Results from a Croatian Tertiary Care Hospital. J Fungi (Basel). 2021 Mar 31;7(4):267. doi: 10.3390/jof7040267. PMID: 33807486; PMCID: PMC8065499.
Comments to the Author
At the same time, it would be important analyze the possibility of placing table 3 in supplementary material. If this is not possible, it would be important to better redistribute column 6 (Candida spp.) So that the words do not are cut off.
Authors' reply
We did all the suggestions and changes proposed by the reviewer. The manuscript is sent with the corrections marked in yellow.
Table 3 was redone in order to be clearly exposed.

Reviewer 2 Report
The manuscript in its current form presents the data on this subject in a cohesive, flowing manner adequately summarizing the current landscape of the research.
Very minor suggestions for further improvement of the manuscript are as follows:
In the prism flow data diagram think about decreasing the size of the yellow oval circle to properly fit around the box that is being addressed. The circle should not be covering any words.
Table 3, while showing dramatic improvement from its former state, still may be difficult to follow due to word wrapping of information in the table. This may again be fixed by changing the size of the font, however I understand there is a lot of data being presented here and this may not be an option.
I would highly suggest looking at the indentation of paragraphs throughout the manuscript. Some paragraphs begin with indentation, while others do not. This is seen in the introduction where none of the paragraphs contain indentation, however as soon as you move into the materials and methods section, indentation appears and remains throughout the rest of the manuscript. However, once you reach the discussion the paragraphs shown do not line up properly. I would just double check the alignment and indentation throughout the manuscript for consistency sake.
Author Response
First of all, I would like to thank the reviewer 2 for the work they have done in correcting our manuscript, which has been very helpful. Below we have answered the questions and addressed the corrections suggested to us:
Reviewer: 2
Comments to the Author
Date of this review
18 Nov 2021 21:00:03
The manuscript in its current form presents the data on this subject in a cohesive, flowing manner adequately summarizing the current landscape of the research.
Very minor suggestions for further improvement of the manuscript are as follows:
In the prism flow data diagram think about decreasing the size of the yellow oval circle to properly fit around the box that is being addressed. The circle should not be covering any words.
Authors' reply
The circle does not exist in the manuscript, it was to indicate that some corrections were made there
Comments to the Author
Table 3, while showing dramatic improvement from its former state, still may be difficult to follow due to word wrapping of information in the table. This may again be fixed by changing the size of the font, however I understand there is a lot of data being presented here and this may not be an option.
I would highly suggest looking at the indentation of paragraphs throughout the manuscript. Some paragraphs begin with indentation, while others do not. This is seen in the introduction where none of the paragraphs contain indentation, however as soon as you move into the materials and methods section, indentation appears and Submission Date
17 October 2021
remains throughout the rest of the manuscript. However, once you reach the discussion the paragraphs shown do not line up properly. I would just double check the alignment and indentation throughout the manuscript for consistency sake.
Authors' reply
We did all the suggestions and changes proposed by the reviewer. The manuscript is sent with the corrections marked in yellow
